# Reliable Active Apprenticeship Learning

**Steve Hanneke**                                          STEVE.HANNEKE@GMAIL.COM
*Purdue University*

**Liu Yang**                                               LIU.YANG0900@OUTLOOK.COM
*Data Intelligence Division, China Unicom Digital Technology Co., Ltd., 133 Xidan North Street*
*Beijing, 100032 China*

**Gongju Wang**                                            WANGGJ129@CHINAUNICOM.CN
*Data Intelligence Division, China Unicom Digital Technology Co., Ltd., 133 Xidan North Street*
*Beijing, 100032 China*

**Yulun Song**                                             SONGYL100@CHINAUNICOM.CN
*Data Intelligence Division, China Unicom Digital Technology Co., Ltd., 133 Xidan North Street*
*Beijing, 100032 China*

**Editors:** Gautam Kamath and Po-Ling Loh

## Abstract

We propose a learning problem, which we call reliable active apprenticeship learning, for which we define a learning algorithm providing optimal performance guarantees, which we further show are sharply characterized by the eluder dimension of a policy class. In this setting, a learning algorithm is tasked with behaving optimally in an unknown environment given by a Markov decision process. The correct actions are specified by an unknown optimal policy in a given policy class. The learner initially does not know the optimal policy, but it has the ability to query an expert, which returns the optimal action for the current state. A learner is said to be reliable if, whenever it takes an action without querying the expert, its action is guaranteed to be optimal. We are then interested in designing a reliable learner which does not query the expert too often. We propose a reliable learning algorithm which provably makes the minimal possible number of queries, which we show is precisely characterized by the eluder dimension of the policy class. We further extend this to allow for imperfect experts, modeled as an oracle with noisy responses. We study two variants of this, inspired by noise conditions from classification: namely, Massart noise and Tsybakov noise. In both cases, we propose a reliable learning strategy which achieves a nearly-minimal number of queries, and prove upper and lower bounds on the optimal number of queries in terms of the noise conditions and the eluder dimension of the policy class.

**Keywords:** Statistical Learning Theory, Apprenticeship Learning, Markov Decision Processes, Reliable Learning, Active Learning, Eluder Dimension

## 1. Introduction

Imagine we design a Mars rover, which drives around on Mars collecting samples and doing various experiments. We want it to be able to drive from place to place autonomously. But we also want it to be careful not to do some suboptimal actions, like driving off a cliff or getting stuck. To avoid this, we will allow it to ask for help sometimes. As it is driving around, when it gets into a state where it is unsure of the optimal action, it sends a request to Earth for a human to take control for a little bit. The human then sends back an optimal action. The rover then follows the given instruction, but also learns from it, so it will better identify the optimal actions in the future. This motivates a problem we call *reliable active apprenticeship learning*.

Formally, we consider a setting, which we call *active apprenticeship learning*. In this setting, there is a state space $\mathcal{S}$, and a set $\mathcal{A}$ of possible actions. There is also an unknown *environment* $P$, defined by several components: (1) an initial state distribution $P_0$, (2) a (time-invariant) state transition distribution $P(\cdot|\cdot)$, specifying the distribution over a next state $s'$ given that action $a$ is taken from current state $s$ by $P(\cdot|s, a)$ (where $s'$ is conditionally independent of any prior history given $s, a$: i.e., a Markov process), and (3) an *oracle* which can (optionally) be *queried* by the learner from its current state $s$ (before the learner chooses its action); when queried, the oracle returns an action $a$ for the learner to take in state $s$. In our most-general results, we allow the oracle to be stochastic, so that its action is sampled from a conditional distribution $P(\cdot|s)$ (and conditionally independent of any prior history given $s$). Thus, an environment $P$ is essentially a Markov decision process (MDP), except that we do not include the traditional notion of *rewards* (as in reinforcement learning) and instead have an oracle which the learner can query from a current state to receive a suggested action.

The learning protocol is defined as follows. The learner first observes a state $s_1$ sampled from the initial state distribution $P_0$. It takes an action $a_1$ from state $s_1$, and then observes a next state $s_2$ sampled from the state transition distribution $P(\cdot|s_1, a_1)$. It then takes an action $a_2$ from $s_2$, and observes a state $s_3$ sampled from $P(\cdot|s_2, a_2)$, and so on, for a number of rounds $T$ (called the *time horizon*). At any time $t$, the learner also has the ability to *query* the oracle from its current state $s_t$, in which case the learner's action $a_t$ follows the oracle's suggested action: i.e., $a_t \sim P(\cdot|s_t)$. So as the learner goes from state to state, sometimes it chooses an action on its own, and sometimes it queries the oracle and follows the oracle's suggested action.

As our objective in the present work, we are interested in learners which have a *reliability* guarantee for their actions. To formalize this, following the traditional abstraction from much of statistical learning theory (Vapnik and Chervonenkis, 1974; Valiant, 1984; Littlestone, 1988), we imagine there is an unknown *optimal policy* $\pi^* : \mathcal{S} \to \mathcal{A}$ which is included in a known (but arbitrary) *policy class* $\Pi \subseteq \mathcal{A}^{\mathcal{S}}$. An active apprenticeship learner is said to be *reliable* if, whenever it does not query the oracle, the action it takes is guaranteed to agree with the optimal policy: that is, $a_t = \pi^*(s_t)$.

Of course, it is easy to define a reliable active apprenticeship learner which simply queries the oracle in every state it encounters. Hence our main question in this work concerns how to make fewer queries while still being reliable. In particular, we aim to identify the *minimum* possible number of queries for reliable active apprenticeship learning: called the *optimal query complexity*.

In this work, we study this problem under increasingly-weaker assumptions on the oracle. Our strongest setting, the *realizable case*, supposes the oracle *always* returns the optimal action $\pi^*(s_t)$ when queried. We sharply characterize the optimal query complexity, which is precisely expressed in terms of the *eluder dimension* of the policy class. We then relax this to merely suppose its distribution $P(\cdot|s_t)$ *favors* the optimal action over other actions, by some margin, analogous to the *Massart noise* assumption in classification or the *gap assumption* in reinforcement learning and contextual bandits. Finally, we further relax this to allow for small margins to occur, though with an assumption that smaller gap sizes occur less frequently, analogous to the *Tsybakov noise* assumption in classification (suitably extended to these MDP-type environments). In both cases, we give general upper and lower bounds on the optimal query complexity, valid for any policy class $\Pi$, expressed in terms of the eluder dimension, along with a learning algorithm which achieves the upper bound without any prior knowledge of the noise conditions.

## 1.1. Related Work

The closely-related subjects of *apprenticeship learning*, *imitation learning*, and *inverse reinforcement learning*, have received much interest in the machine learning literature (e.g., Ross, Gordon, and Bagnell, 2011; Abbeel and Ng, 2004; Ng and Russell, 2000). These settings are based on a similar setup to ours, in that the learner observes an oracle's behavior in the Markov decision process. However, they differ in that they are *episodic*, with the oracle demonstrating the ideal behavior for several *entire episodes*, and the interest is in learning how to mimic the oracle's behavior in *future episodes*. In contrast, our setting consists of a *single* episode, and we are tasked with learning how to mimic the ideal behavior *on-the-fly*. Unlike those settings, we also require a *reliability* guarantee for the learner, rather than considering quantitative notions of performance, such as by cumulative *rewards*.

A recent work of Sekhari, Sridharan, Sun, and Wu (2023) considers an active imitation learning setting, which closely matches our setting, in that the goal is to compete with an optimal policy $\pi^*$, and the learner may query an oracle to receive (noisy) samples of the $\pi^*$ actions. However, their setting differs from ours in several important respects. For one, they study an *episodic* learning problem, where their learner is aided by the independence across episodes. Additionally, rather than a policy class $\Pi$, they directly model the oracle's action distribution $P(\cdot|s)$ as a composition of an unknown function $f^*$ from a known online-learnable class $\mathcal{F}$ of regression functions, with a strongly-convex link function. In contrast, we allow oracles $P(\cdot|s)$ merely satisfying general noise conditions (analogous to Massart or Tsybakov noise conditions from classification); in particular, in our setting, the set of admitted oracle response distributions $P(\cdot|s)$ is *not* estimable from samples. Finally, our additional requirement that the learner be *reliable* makes the learning problem we consider significantly more challenging for the learner. Nonetheless, it is interesting that their guarantees are also expressed in terms of the eluder dimension (in their case, it is the eluder dimension of the function class).

The general subject of *reliable* learning was first studied in the classification setting with i.i.d. samples by Rivest and Sloan (1988). In that literature, the learners predictions are required to match an unknown *target concept*, and for examples where this cannot be guaranteed, the learner is permitted to *abstain* from predicting. The interest is then in understanding the number of samples sufficient to guarantee that the frequency of abstention is small. This idea was further studied in an i.i.d. sequential prediction setting by El-Yaniv and Wiener (2010) (termed *perfect selective classification*), and extended to allow for non-realizable queries by El-Yaniv and Wiener (2011); Gelbhart and El-Yaniv (2019).

The theory we develop, on reliable active apprenticeship learning, is aided by an important and well-known connection between reliable learning and *active learning*. Specifically, the core technique underlying our algorithms and analysis below, and common to both the reliable classification literature and the active learning literature for classification, is based on the principle of *disagreement-based learning*. This approach maintains a set of *surviving* policies $V \subseteq \Pi$ up to each round $t$, and only declares the optimal action from $s_t$ as *certain* if *all* policies in $V$ *agree* on what action to take from $s_t$. As long as our updates to $V$ in each round retain $\pi^* \in V$, this clearly specifies a reliable learner, so that the main interest in this approach is in defining updates to $V$ which most-rapidly reduce the *region of disagreement* $\{s \in \mathcal{S} : \exists \pi, \pi' \in V, \pi(s) \neq \pi'(s)\}$ while retaining $\pi^* \in V$. Disagreement-based learning was introduced by Cohn, Atlas, and Ladner (1994) in the context of realizable-case active learning, and has been extended to allow for various

non-realizable settings in a rich and well-developed literature on the theory of active learning (see e.g. Balcan, Beygelzimer, and Langford, 2009; Hanneke, 2007, 2014; Dasgupta, Hsu, and Monteleoni, 2007). It has also been used to understand reliable *prediction* in i.i.d. classification settings (Rivest and Sloan, 1988; El-Yaniv and Wiener, 2010, 2011; Balcan, Blum, Hanneke, and Sharma, 2022; Balcan, Hanneke, Pukdee, and Sharma, 2023), and to study the optimal *regret* achievable in stochastic *contextual bandits* and *reinforcement learning* (Foster, Rakhlin, Simchi-Levi, and Xu, 2021). Since our setting lies at the intersection of reliable learning and active learning, both of which have disagreement-based learning as a core principle, disagreement-based learning is a particularly natural principle on which to base our theory, and moreover, as we argue below, leads to *optimal* query complexities. To our knowledge, our work is the first to study reliable learning in MDP-type environments, and to formulate the general problem of reliable active apprenticeship learning.

## 2. Summary of Main Results

In this section, we introduce key definitions and summarize the main results of this work.

**Notation:** Let us first introduce a bit of useful terminology and notation (used in the algorithms and proofs below). For any $V \subseteq \Pi$, define the *region of disagreement*:

$$\text{DIS}(V) = \{s \in \mathcal{S} : \exists \pi, \pi' \in V, \pi(s) \neq \pi'(s)\}.$$

Additionally, we refer to any sequence $(s_1, a_1), \ldots, (s_T, a_T)$ in $\mathcal{S} \times \mathcal{A}$ as a *trajectory*. In particular, for any environment $P$ and any active apprenticeship learner $\mathbb{A}$, we will refer to the (possibly stochastic) sequence of states and actions $(s_1, a_1), \ldots, (s_T, a_T)$ followed by the learner as the *learner's trajectory*.

**The Eluder Dimension:** The results in this work will hold for *any* state space $\mathcal{S}$, action set $\mathcal{A}$, time horizon $T$, and policy class $\Pi$. As such, naturally, the value of the optimal query complexity will be informed by the specific policy class $\Pi$. We will argue that, under several different noise models for the oracle, the dependence on $\Pi$ in the optimal query complexity is precisely captured by the *eluder dimension* of the policy class $\Pi$. Formally, the eluder dimension, as originally introduced by Russo and Van Roy (2013); Osband and Van Roy (2014); Foster, Rakhlin, Simchi-Levi, and Xu (2021) (more specifically termed the *policy* eluder dimension in those works), is defined as follows.

**Definition 1** *For a given policy $\pi_0$, the* eluder dimension $\mathfrak{e}_{\pi_0} := \mathfrak{e}_{\pi_0}(\Pi)$ *is defined as the largest* $n \in \mathbb{N}$ *such that* $\exists s_1, \ldots, s_n \in \mathcal{S}$ *such that*

$$\forall i \leq n, s_i \in \text{DIS}(\{\pi \in \Pi : \forall j < i, \pi(s_j) = \pi_0(s_j)\}).$$

*Any such sequence is called an* eluder sequence *centered at $\pi_0$. If no such largest $n$ exists, define* $\mathfrak{e}_{\pi_0} = \infty$. *Also define* $\mathfrak{e}(\Pi) := \sup_{\pi_0 \in \Pi} \mathfrak{e}_{\pi_0}$.

The eluder dimension (and its variants for $\mathbb{R}$-valued functions) has been a useful quantity in the literature on reinforcement learning and contextual bandits (Russo and Van Roy, 2013; Osband and Van Roy, 2014; Foster, Rakhlin, Simchi-Levi, and Xu, 2021) and has interesting relations to other important quantities in learning theory (Li, Kamath, Foster, and Srebro, 2022; Hanneke, 2024).

**The Realizable Case:** We begin with the simplest variant of our setting, corresponding to the strongest assumption on the oracle considered in this work. Later results below will considerably relax this assumption. Formally, we say the environment $P$ has a *realizable oracle*[1] with optimal policy $\pi^*$ if, for any state $s$, querying the oracle from state $s$ *always* returns $\pi^*(s)$.

**Definition 2** *An active apprenticeship learner $\mathbb{A}$ is said to be* reliable *for the* realizable case *if, for every choice of $\pi^* \in \Pi$, for every environment $P$ with a realizable oracle with optimal policy $\pi^*$, the learner's trajectory $(s_1, a_1), \ldots, (s_T, a_T)$ always* satisfies that every state $s_t$ in which the learner does not query the oracle has $a_t = \pi^*(s_t)$.

By definition of the protocol, every state $s_t$ in which the learner *does* query the oracle necessarily satisfies $a_t = \pi^*(s_t)$ as well. Thus, a reliable active apprenticeship learner for the realizable case always has a trajectory $(s_1, a_1), \ldots, (s_T, a_T)$ equal $(s_1, \pi^*(s_1)), \ldots, (s_T, \pi^*(s_T))$ under a realizable oracle: that is, it exactly mimics the optimal policy. Our first result sharply characterizes the optimal query complexity of reliable active apprenticeship learning in the realizable case.

**Theorem 3 (Realizable-case Optimal Query Complexity)** *There is a reliable active apprenticeship learner $\mathbb{A}_{\mathrm{CAL}}$ for the realizable case which, for every $\pi^* \in \Pi$, for every environment $P$ with a realizable oracle with optimal policy $\pi^*$, the number of queries by $\mathbb{A}_{\mathrm{CAL}}$ is at most $\min\{\mathfrak{e}_{\pi^*}, T\}$. Moreover, for* any *reliable active apprenticeship learner $\mathbb{A}$ for the realizable case, for every $\pi^* \in \Pi$, there exists a (deterministic) environment $P$ with a realizable oracle with optimal policy $\pi^*$ such that the number of queries by $\mathbb{A}$ is at least $\min\{\mathfrak{e}_{\pi^*}, T\}$.*
*Thus, for every $\pi^* \in \Pi$, the (minimax) optimal query complexity of reliable active apprenticeship learning in the realizable case with optimal policy $\pi^*$ is precisely $\min\{\mathfrak{e}_{\pi^*}, T\}$.*

The algorithm $\mathbb{A}_{\mathrm{CAL}}$ is specifically based on a well-known strategy from the active learning literature, known as *disagreement-based* learning, introduced by Cohn, Atlas, and Ladner (1994). Specifically, it is defined as follows.

---

Algorithm $\mathbb{A}_{\mathrm{CAL}}$:
Initialize $V = \Pi$
For each time $t = 1, 2, \ldots, T$,
    If $s_t \in \mathrm{DIS}(V)$
        Query the oracle to receive action $a_t = \pi^*(s_t)$ and take action $\hat{a}_t = a_t$
        Update $V \leftarrow \{\pi \in V : \pi(s_t) = \pi^*(s_t)\}$
    Else take the unique action $\hat{a}_t$ in $\{\pi(s_t) : \pi \in V\}$

---

We present the proof of Theorem 3, establishing an upper bound for $\mathbb{A}_{\mathrm{CAL}}$ and a matching minimax lower bound, in Section 3 below.

---

1. The term "realizable" has been overloaded in the literature. Here we adopt the usage from the classification literature. In the literature on contextual bandits and reinforcement learning, the term is sometimes used instead for what is historically known as a *well-specified model* assumption: that is, that an optimal function is contained in a given function class, even if there is also some noise.

**Preferential Noisy Oracles:** In realistic scenarios, the realizability assumption is quite strong. For instance, it does not allow for somewhat ambiguous scenarios, where the action a human expert would recommend may vary from the $\pi^*$ action.[2] To allow for such possibilities, it makes sense to relax the realizability assumption. In this work, we model this aspect by considering oracles with *stochastic* responses. However, to retain a well-defined notion of *reliability*, we continue to suppose there is a fixed optimal policy $\pi^* \in \Pi$, which the adversary *tends to prefer*. Specifically, we consider the following definition.

**Definition 4** *We say the environment $P$ has a* preferential noisy oracle *with optimal policy $\pi^*$ if, for any state $s$, querying the oracle from state $s$ returns a sample $a \sim P(\cdot|s)$ (conditionally independent of the history given $s$), where the conditional distribution $P(\cdot|s)$ satisfies that $P(\pi^*(s)|s) > \max_{a \neq \pi^*(s)} P(a|s)$.*

Since oracles of this type have stochastic responses, in order to achieve non-trivial query complexity, we will need to mildly relax the reliability requirement to allow for a small failure probability. For this purpose, let us fix some $\delta \in (0, 1)$, and we will require the learner to satisfy the reliability guarantee with probability at least $1 - \delta$.[3] Formally, we adopt the following definition.

**Definition 5** *An active apprenticeship learner $\mathbb{A}$ is said to be* reliable *for the* preferential case *if, for every $\pi^* \in \Pi$, for every environment $P$ with a preferential noisy oracle with optimal policy $\pi^*$, with probability at least $1 - \delta$, the learner's trajectory $(s_1, a_1), \ldots, (s_T, a_T)$ satisfies that every state $s_t$ in which the learner does* not *query the oracle has $a_t = \pi^*(s_t)$.*

Following a common pattern from the active learning literature (Balcan, Beygelzimer, and Langford, 2006; Dasgupta, Hsu, and Monteleoni, 2007; Hanneke, 2007, 2014), we generalize the $\mathbb{A}_{\text{CAL}}$ algorithm to remain reliable under preferential noisy oracles. Throughout the remainder of the paper, for simplicity we denote by $\log(x) = \log_2(\max\{x, 2\})$. Fix a sufficiently large universal constant $c > 0$ (informed by the proofs below). Also, for any $t \in \mathbb{N}$, define $\delta_t = \frac{\delta}{2(t+1)^2|\Pi|^2}$.[4]

---

2. For instance, in our Mars rover example from Section 1, when navigating to avoid a rock, it may be reasonable to drive around either to the left or right of it; while one option may be slightly better than the other, a human expert might make such suggestions quickly, and hence have some probability of suggesting the slightly-less-optimal action.

3. The necessity of allowing this $\delta$ failure probability follows from our lower bound in Theorem 9.

4. For simplicity, we have stated the algorithm and theorems expressed in terms of $|\Pi|$, the size of the policy class. While it is straightforward, using standard techniques, to replace this with instead a dependence on the *sequential graph dimension* of the policy class (Hanneke, Moran, Raman, Subedi, and Tewari, 2023), we remark that a recent work of Hanneke (2024) shows that we always have $\mathfrak{e}(\Pi) = \Omega(\log_{|\mathcal{A}|}(|\Pi|))$, and thus a dependence on $|\Pi|$ is unavoidable in the results (for finite $\mathcal{A}$), so that the $\log(|\Pi|)$ appearing in our bounds may be considered an inherent dependence anyway.

Algorithm: ReliableApprentice
0. Initialize $V = \Pi$, $Q = \{\}$; let $s_1$ be the initial state sampled from $P_0$
1. For $t = 1, 2, \ldots, T$
2.    If $s_t \in \mathrm{DIS}(V)$
3.       Query for $a_t \sim P(\cdot|s_t)$ and take action $\hat{a}_t = a_t$
4.       Update $Q \leftarrow Q \cup \{(s_t, a_t)\}$
5.       Let $\hat{\pi}_t = \mathrm{argmax}_{\pi \in V} \sum_{(s,a) \in Q} \mathbb{1}[\pi(s) = a]$
6.       Update

$$V \leftarrow \left\{ \pi \in V : \sum_{(s,a) \in Q} (\mathbb{1}[\hat{\pi}_t(s) = a] - \mathbb{1}[\pi(s) = a]) \leq \right.$$
$$\left. c\sqrt{\left( \sum_{(s,a) \in Q} \mathbb{1}[\{a\} \subsetneq \{\pi(s), \hat{\pi}_t(s)\}] \right) \log\left(\frac{1}{\delta_t}\right) + c\log\left(\frac{1}{\delta_t}\right)} \right\}$$

7.    Else take the unique action $\hat{a}_t$ in $\{\pi(s_t) : \pi \in V\}$

As the following theorem establishes that this algorithm is indeed reliable in the preferential case. Its proof is included in Section 4 below.

**Theorem 6** *The algorithm* ReliableApprentice *is reliable for the preferential case.*

To state quantitative query complexity bounds under preferential noisy oracles, we introduce special cases of preferential noisy oracles, inspired by commonly-studied noise models from the classification literature.

**Massart Noise:** As a first generalization beyond the realizable case, we follow a common idea from the literature on classification (Massart and Nédélec, 2006) (similar ideas have arisen in the literature on contextual bandits and reinforcement learning Foster, Rakhlin, Simchi-Levi, and Xu, 2021). Specifically, we suppose that in each state $s$, the oracle's response distribution $P(\cdot|s)$ prefers the optimal $\pi^*(s)$ action with a probability *bounded away* from the probabilities of the other actions. This is often referred to as a *gap* assumption or *bounded noise* assumption in the statistical learning theory literature, and is also often referred to as *Massart noise* in honor of the seminal analysis of Massart (2007); Massart and Nédélec (2006) under this assumption. Formally, we have the following definition.

**Definition 7** *For any $\Delta \in (0, 1]$, we say an environment $P$ satisfies the* Massart noise *condition with optimal policy $\pi^* \in \Pi$ and gap $\Delta$ if, for any state $s$, the oracle's response distribution to queries from state $s$ satisfies*

$$P(\pi^*(s)|s) \geq \max_{a \neq \pi^*(s)} P(a|s) + \Delta.$$

We will establish upper and lower bounds for reliable active apprenticeship learning with Massart noise. Formally, we prove the following results. Their proofs are presented in Section 5 below.

**Theorem 8** *For any $\Delta \in (0, 1]$, for any $\pi^* \in \Pi$, and any environment $P$ satisfying the Massart noise condition (Definition 7) with optimal policy $\pi^*$ and gap $\Delta$, with probability at least $1 - \delta$, the algorithm* ReliableApprentice *(already shown to be reliable in Theorem 6) makes a number of queries at most*

$$O\left( \mathfrak{e}_{\pi^*} \frac{1}{\Delta^2} \log\left( \frac{|\Pi|T}{\delta} \right) \right).$$

We complement this with the following lower bound, revealing that, again, the optimal query complexity is captured by the eluder dimension (even when allowing for a $\delta$ failure probability).

**Theorem 9** *Suppose $|\Pi| \geq 2$. For any $\Delta \in (0, 1/8)$, if $\delta \in (0, 1/16e)$, for any $\pi^* \in \Pi$, there is an environment $P$ satisfying the Massart noise condition with optimal policy $\pi^*$ and gap $\Delta$ such that, for any active apprenticeship learning algorithm $\mathbb{A}$ which is reliable under Massart noise, with probability greater than $\delta$ the algorithm makes a number of queries at least*

$$\Omega\left( \left( \mathfrak{e}_{\pi^*} + \frac{1}{\Delta^2} \log\left( \frac{1}{\delta} \right) \right) \wedge T \right).$$

**The Mixed-Margin Condition, a Generalization of Tsybakov Noise for MDP Environments:** In considering extensions beyond the Massart noise assumption, we may again take inspiration from the classification literature. In the context of learning a classifier from i.i.d. samples, a condition known as the *margin condition* (and often referred to as *Tsybakov noise*, in honor of the seminal analysis of Mammen and Tsybakov, 1999; Tsybakov, 2004 under this condition) extends the Massart noise assumption by allowing for some states to have smaller gaps $\Delta$ than others, as long as such small-gap states do not occur too frequently: the smaller the gap, the less frequently they occur, and the noise model expresses this relation as being related polynomially with parameters $(C, \alpha)$ expressing this relation. However, we face a considerable challenge in appropriately formulating such a condition in our setting, since (unlike classification under i.i.d. samples) it is less clear what choice of measure is appropriate for defining what we mean by "less frequently". One of the contributions of the present work is formulating an appropriate definition of the margin condition, in the context of reliable apprenticeship learning in MDP-type environments. The key insight is that the condition need only control the frequency of small-gap states among trajectories which a reliable active apprenticeship learner may potentially follow: namely, *mixed optimal trajectories*.

Formally, for any environment $P$ with a preferential noisy oracle with optimal policy $\pi^*$, we say an $(\mathcal{S} \times \mathcal{A})^T$-valued random sequence $(\mathbf{s}_1, \mathbf{a}_1, \ldots, \mathbf{s}_T, \mathbf{a}_T)$ is a *mixed optimal trajectory* if (1) it is a trajectory in the environment $P$ (i.e., $\mathbf{s}_1$ has the initial state distribution, and each $\mathbf{s}_{t+1}$ has conditional distribution $P(\cdot | \mathbf{s}_t, \mathbf{a}_t)$ for each $t < T$, and is conditionally independent of $\{\mathbf{s}_i, \mathbf{a}_i\}_{i < t}$ given $\mathbf{s}_t, \mathbf{a}_t$), and (2) for each $t \leq T$, either $\mathbf{a}_t = \pi^*(\mathbf{s}_t)$ or $\mathbf{a}_t$ has conditional distribution $P(\cdot | \mathbf{s}_t)$ given $\mathbf{s}_t$ (and is conditionally independent of $\{\mathbf{s}_i, \mathbf{a}_i\}_{i < t}$ given $\mathbf{s}_t$): i.e., the distribution of an oracle query. In particular, by definition, any reliable learner follows a mixed optimal trajectory with probability at least $1 - \delta$.

Since mixed optimal trajectories are themselves stochastic (since even the oracle's responses may be stochastic), we will need to express the condition on the frequency of small-gap states as a probabilistic inequality. Specifically, we propose the following extension of the classic *Tsybakov noise* condition (Mammen and Tsybakov, 1999; Tsybakov, 2004) to MDP-type environments under mixed optimal trajectories.

**Definition 10** *We say an environment $P$ with preferential noisy oracle with optimal policy $\pi^* \in \Pi$ satisfies the* mixed margin condition *with parameters $(C, \alpha) \in [1, \infty) \times [0, 1)$ if for every $\delta' \in (0, 1)$, for every $t \le T$, for every mixed optimal trajectory $(\mathbf{s}_1, \mathbf{a}_1, \dots, \mathbf{s}_T, \mathbf{a}_T)$, with probability at least $1 - \delta'$, every $\tau \in (0, 1)$ satisfies*

$$\frac{1}{t} \sum_{t'=1}^{t} \mathbb{1} \left[ P(\pi^*(\mathbf{s}_{t'}) | \mathbf{s}_{t'}) - \max_{a \neq \pi^*(\mathbf{s}_{t'})} P(a | \mathbf{s}_{t'}) \le \tau \right] \le C \tau^{\frac{\alpha}{1-\alpha}} + \frac{1}{t} \log\left(\frac{1}{\delta'}\right).$$

We establish upper and lower bounds on the optimal query complexty of reliable active apprenticeship learning under the mixed margin condition, stated in the following theorems. Their proofs are presented in Section 6 below.

**Theorem 11** *For any $P$ satisfying the mixed margin condition (Definition 10) with parameters $(C, \alpha)$ and optimal policy $\pi^* \in \Pi$, with probability at least $1 - 2\delta$, the algorithm $\mathrm{ReliableApprentice}$ (already shown to be reliable in Theorem 6) makes a number of queries at most*

$$O\left( \mathfrak{e}_{\pi^*} T^{\frac{2-2\alpha}{2-\alpha}} \left( \log\left(\frac{|\Pi|T}{\delta}\right) \right)^{\frac{\alpha}{2-\alpha}} \right).$$

This upper bound is complemented by the following lower bound.

**Theorem 12** *Fix any $\pi^* \in \Pi$ and $(C, \alpha) \in [64, \infty) \times (0, 1)$ s.t. $T \ge 64 \cdot \max\left\{ (5/2)^{\frac{2-\alpha}{1-\alpha}}, 16^{\frac{2-\alpha}{\alpha}} \right\}$. Suppose $\delta$ is upper bounded by a sufficiently small universal constant (discussed in the proof). Suppose $\Pi$ satisfies a* non-triviality *condition: $\exists s_0, s_1$ such that $\exists \pi_1 \in \Pi$ with $\pi_1(s_0) = \pi^*(s_0)$ and $\pi_1(s_1) \neq \pi^*(s_1)$. There exists an environment $P$ satisfying the mixed margin condition with parameters $(C, \alpha)$ and optimal policy $\pi^*$ such that, for any active apprenticeship learner reliable under the mixed margin condition (with parameters $(C, \alpha)$), with probability at least $(1 - 2e^{-1}) \frac{1}{16e}$, its number of queries is at least*

$$\Omega\left( \left( \mathfrak{e}_{\pi^*} + T^{\frac{2-2\alpha}{2-\alpha}} \right) \wedge T \right).$$

**Outline of the paper:** The rest of the paper provides detailed proofs of these results. The results on the realizable case are presented in Section 3, followed by the Massart noise case in Section 5, and the results for Tsybakov noise in Section 6. We conclude with future directions and open questions (e.g., extension to the *agnostic case*) in Section 7.

## 3. Realizable Case

This section presents the proof of Theorem 3, establishing the optimal query complexity of reliable active apprenticeship learning in the realizable case.

**Proof of Theorem 3** We begin with the positive results for $\mathbb{A}_{\mathrm{CAL}}$. First note that, since the algorithm only updates $V$ by constraining to agree with $\pi^*$ on a state $s_t$ it has queried, it trivially satisfies that $\pi^* \in V$ is maintained as an invariant. In particular, since any state $s_t$ it does not query satisfies $s_t \notin \mathrm{DIS}(V)$, it follows that the unique action $\hat{a}_t$ agreed upon by all policies in $V$ must be the action $\pi^*(s_t)$. Hence $\mathbb{A}_{\mathrm{CAL}}$ is indeed reliable in the realizable case. Moreover, note that the

update to $V$ ensures that all policies retained in $V$ after a query agree with $\pi^*(s_t)$ on the state $s_t$. Since all policies in $V$ trivially agree with $\pi^*(s_t)$ on rounds where it does *not* query, together we have that all policies in $V$ agree on *all* past states, queried or unqueried: that is, after each round $t$, we have that $V = \{\pi \in \Pi : \forall t' \leq t, \pi(s_{t'}) = \pi^*(s_{t'})\}$.

Next we upper bound the number of queries. Let $(s_1, \hat{a}_1), \ldots, (s_T, \hat{a}_T)$ be the trajectory followed by $\mathbb{A}_{\mathrm{CAL}}$, and let $t_1, \ldots, t_n$ be the subsequence of all $t \in \{1, \ldots, T\}$ for which

$$s_t \in \mathrm{DIS}(\{\pi \in \Pi : \forall t' < t, \pi(s_{t'}) = \pi^*(s_{t'})\}).$$

Note that this is precisely the subsequence of times where the algorithm queries the oracle. Since the region of disagreement is non-decreasing in its argument set, this also implies

$$\forall i \leq n, s_{t_i} \in \mathrm{DIS}(\{\pi \in \Pi : \forall j < i, \pi(s_{t_j}) = \pi^*(s_{t_j})\}).$$

Note that this precisely matches the definition of an eluder sequence centered at $\pi^*$, and thus $s_{t_1}, \ldots, s_{t_n}$ witness the fact that $\mathfrak{e}_{\pi^*} \geq n$ in Definition 1. Therefore, the total number $n$ of queries satisfies $n \leq \mathfrak{e}_{\pi^*}$, which completes the proof of the upper bound on the number of queries (noting that $T$ is always trivially an upper bound).

Next we prove this is also a lower bound (following an argument similar in spirit to one from perfect selective classification, El-Yaniv and Wiener, 2010, modified to fit our setting). Let $s_1, \ldots, s_n$ be an eluder sequence centered at $\pi^*$ (as defined in Definition 1), for finite $n \leq \min\{\mathfrak{e}_{\pi^*}, T\}$, for a given $\pi^* \in \Pi$. If $n < T$, also extend the sequence to any $s_{n+1}, \ldots, s_T$ all equal $s_n$, to define a complete state sequence. Note that, from its definition, it must be that the $s_1, \ldots, s_n$ states are all distinct. Define an environment $P$ with deterministic state sequence $s_i$, $i = 1, \ldots, T$ (i.e., $P_0(s_1) = 1$, and for $1 \leq i < n$, for any $a \in \mathcal{A}$ the transition probabilities satisfy $P(s_{i+1}|s_i, a) = 1$, and $P(s_n|s_n, a) = 1$). Also define an oracle which deterministically returns $\pi^*(s)$ from any state $s$, so that the environment $P$ indeed satisfies the realizable case with optimal policy $\pi^*$.

By definition, each $i \leq n$ has

$$s_i \in \mathrm{DIS}(\{\pi \in \Pi : \forall j < i, \pi(s_j) = \pi^*(s_j)\}). \tag{1}$$

Consider any active apprenticeship learner $\mathbb{A}$ for which, in the above environment $P$, the algorithm has a non-zero probability (allowing it possibly to be randomized) of *not* querying in some state $s_t$ in the sequence, $t \leq n$, and suppose $t$ is the smallest such index for which it has a non-zero probability. There are now two cases to consider. First, suppose the algorithm has a non-zero probability of taking an action $a \neq \pi^*(s_t)$ in state $s_t$. In this case, by definition, the algorithm cannot be reliable, as witnessed by having this non-zero probability of taking an action other than $\pi^*(s_t)$ in state $s_t$. On the other hand, consider the case that the algorithm still has a non-zero probability of not querying in state $s_t$ (where $t$ is smallest with this property), and yet it has probability one of taking action $a_t = \pi^*(s_t)$ under $P$. Consider an *alternative* realizable-case environment $P_t$, which has identical initial state distribution and state transitions to $P$, but has optimal policy $\pi_t \in \Pi$ such that $\pi_t(s_i) = \pi^*(s_i)$ for all $i < t$, and $\pi_t(s_t) \neq \pi^*(s_t)$. Such a policy $\pi_t$ must exist, by the defining property (1) of the $s_i$ sequence. Note that the distribution of actions, states, and query responses up until arriving in state $s_t$ at time $t$ are identical under $P$ and $P_t$, so under $P_t$ the algorithm still has a non-zero probability of not querying in state $s_t$ and still has conditional probability *one* of taking action $\pi^*(s_t)$ given that it does not query in state $s_t$. Thus, since $\pi_t(s_t) \neq \pi^*(s_t)$, we have that under $P_t$ there is a non-zero probability that the algorithm does not query $s_t$ and yet takes action

$\pi^*(s_t)$, which is not the optimal action in state $s_t$ under environment $P_t$. Thus, such an algorithm cannot be reliable in the realizable case. Altogether, we conclude that any reliable algorithm must have probability one of querying all of $s_1, \ldots, s_n$ when run under environment $P$. This completes the proof. ∎

## 4. Reliable Learning Under Preferential Noisy Oracles

We now present the proof of Theorem 6, establishing that ReliableApprentice is reliable in the preferential case.

**Proof of Theorem 6** The theorem will follow from the following claim: on an event $E_1$ of probability at least $1 - \delta$, $\pi^* \in V$ is maintained as an invariant on all rounds of ReliableApprentice. This will follow from a martingale uniform concentration inequality. Let $I_t \in \{0, 1\}$ be 1 iff $s_t \in \mathrm{DIS}(V)$ on round $t$. For any $\pi, \pi' \in \Pi$, the sequence

$$\sum_{t'=1}^{t} \left( \mathbb{1}[\pi(s_{t'}) = \hat{a}_{t'}] - \mathbb{1}[\pi'(s_{t'}) = \hat{a}_{t'}] \right) I_{t'}$$
$$- \mathbb{E}\left[ \sum_{t'=1}^{t} \left( \mathbb{1}[\pi(s_{t'}) = \hat{a}_{t'}] - \mathbb{1}[\pi'(s_{t'}) = \hat{a}_{t'}] \right) I_{t'} \middle| s_1, \ldots, s_t, \hat{a}_1, \ldots, \hat{a}_{t-1} \right]$$

is a martingale difference sequence with respect to $(s_1, \ldots, s_t, \hat{a}_1, \ldots, \hat{a}_{t-1})$. Noting that

$$\left( \left( \mathbb{1}[\pi(s_{t'}) = \hat{a}_{t'}] - \mathbb{1}[\pi'(s_{t'}) = \hat{a}_{t'}] \right) I_{t'} \right)^2 = \mathbb{1}\left[ \{\hat{a}_{t'}\} \subsetneq \{\pi(s_{t'}), \pi'(s_{t'})\} \right] I_{t'},$$

the empirical Bernstein inequality for martingale differences (Bernstein, 1927) implies that, with probability at least $1 - \delta_t$,

$$\left| \sum_{t'=1}^{t} \left( \mathbb{1}[\pi(s_{t'}) = \hat{a}_{t'}] - \mathbb{1}[\pi'(s_{t'}) = \hat{a}_{t'}] \right) I_{t'} \right.$$
$$\left. - \sum_{t'=1}^{t} \mathbb{E}\left[ \left( \mathbb{1}[\pi(s_{t'}) = \hat{a}_{t'}] - \mathbb{1}[\pi'(s_{t'}) = \hat{a}_{t'}] \right) I_{t'} \middle| s_1, \ldots, s_{t'}, \hat{a}_1, \ldots, \hat{a}_{t'-1} \right] \right|$$
$$\leq c_0 \sqrt{ \left( \sum_{t'=1}^{t} \mathbb{1}[\{\hat{a}_{t'}\} \subsetneq \{\pi(s_{t'}), \pi'(s_{t'})\}] I_{t'} \right) \log\left( \frac{1}{\delta_t} \right) } + c_0 \log\left( \frac{1}{\delta_t} \right) \quad (2)$$

for an appropriate universal constant $c_0$. Recalling that $\delta_t \leq \frac{\delta}{(t+1)^2 |\Pi|^2}$, by the union bound we have that the inequality above holds simultaneously for all $t \leq T$ and $\pi, \pi' \in \Pi$ with probability at least $1 - \delta$. Denote this event as $E_1$.

We are now ready to establish the claim that $\pi^* \in V$ is maintained as an invariant. Since this is satisfied at the start of round 1, we may take this as a base case in an inductive argument. For the purpose of induction, suppose $\pi^* \in V$ at the start of round $t$. If $s_t \notin \mathrm{DIS}(V)$ on round $t$, then $V$ is not changed on round $t$ and the invariant $\pi^* \in V$ is maintained trivially. Otherwise, suppose

$s_t \in \text{DIS}(V)$, and that the event $E_1$ holds. Since $\hat{\pi}_t \in V$ as well, applying the above inequality (2) with $\pi = \hat{\pi}_t$ and $\pi' = \pi^*$, we have that

$$\sum_{t'=1}^{t} \left( \mathbb{1}[\hat{\pi}_t(s_{t'}) = \hat{a}_{t'}] - \mathbb{1}[\pi^*(s_{t'}) = \hat{a}_{t'}] \right) I_{t'}$$

$$\leq \sum_{t'=1}^{t} \mathbb{E}[(\mathbb{1}[\hat{\pi}_t(s_{t'}) = \hat{a}_{t'}] - \mathbb{1}[\pi^*(s_{t'}) = \hat{a}_{t'}]) I_{t'} | s_1, \ldots, s_{t'}, \hat{a}_1, \ldots, \hat{a}_{t'-1}]$$

$$+ c_0 \sqrt{\left( \sum_{t'=1}^{t} \mathbb{1}[\{\hat{a}_{t'}\} \subsetneq \{\hat{\pi}_t(s_{t'}), \pi^*(s_{t'})\}] I_{t'} \right) \log\left(\frac{1}{\delta_t}\right)} + c_0 \log\left(\frac{1}{\delta_t}\right).$$

Since we generally have

$$\mathbb{E}[(\mathbb{1}[\hat{\pi}_t(s_{t'}) = \hat{a}_{t'}] - \mathbb{1}[\pi^*(s_{t'}) = \hat{a}_{t'}]) I_{t'} | s_1, \ldots, s_{t'}, \hat{a}_1, \ldots, \hat{a}_{t'-1}]$$
$$= (P(\hat{\pi}_t(s_{t'})|s_{t'}) - P(\pi^*(s_{t'})|s_{t'})) I_{t'} \leq 0$$

by the preferential noise assumption (recall Definition 4), we conclude that

$$\sum_{t'=1}^{t} \left( \mathbb{1}[\hat{\pi}_t(s_{t'}) = \hat{a}_{t'}] - \mathbb{1}[\pi^*(s_{t'}) = \hat{a}_{t'}] \right) I_{t'}$$

$$\leq c_0 \sqrt{\left( \sum_{t'=1}^{t} \mathbb{1}[\{\hat{a}_{t'}\} \subsetneq \{\hat{\pi}_t(s_{t'}), \pi^*(s_{t'})\}] I_{t'} \right) \log\left(\frac{1}{\delta_t}\right)} + c_0 \log\left(\frac{1}{\delta_t}\right).$$

Thus, $\pi^*$ will be preserved in $V$ at the end of round $t$, for an appropriate choice $c = c_0$ of the universal constant $c$. This establishes the invariant that $\pi^* \in V$ on all rounds, on the event $E_1$, by the principle of induction. In particular, since the algorithm queries every $s_t \in \text{DIS}(V)$, we may conclude that on every round the algorithm does not query $s_t$, the action $\hat{a}_t$ is agreed upon by all $\pi \in V$ at that time, and since $\pi^* \in V$, this implies this action satisfies $\hat{a}_t = \pi^*(s_t)$. Thus, on the event $E_1$, the algorithm's actions are reliable, in the sense required by Definition 5. ∎

## 5. Massart Noise

This section presents the proofs of Theorems 8 and 9, establishing query complexity upper and lower bounds under Massart noise. One key fact we will need for establishing Theorem 8 is the following combinatorial lemma.

**Lemma 13** *Fix any $n \in \mathbb{N}$, $k \in \mathbb{N} \cup \{0\}$, and any $\pi_0 \in \Pi$, and let $s_1, \ldots, s_n \in \mathcal{S}$ satisfy that, $\forall i \in \{1, \ldots, n\}$,*

$$s_i \in \text{DIS}\left( \left\{ \pi \in \Pi : \sum_{t<i} \mathbb{1}[\pi(s_t) \neq \pi_0(s_t)] \leq k \right\} \right). \tag{3}$$

*Then there exists $m \in \mathbb{N}$ with $m \geq \frac{n}{k+1}$, and $i_1 < \cdots < i_m$ in $\{1, \ldots, n\}$, s.t. $\forall j \in \{1, \ldots, m\}$,*

$$s_{i_j} \in \text{DIS}(\{\pi \in \Pi : \forall t < j, \pi(s_{i_t}) = \pi_0(s_{i_t})\}). \tag{4}$$

*In particular, this implies $n \leq (k+1)\mathfrak{e}_{\pi_0}$.*

**Proof** We proceed by induction on $n$, for any fixed value $k \in \mathbb{N} \cup \{0\}$. If $n \leq k+1$, the property (3) for $i = 1$ already implies (4) holds for the subsequence $s_1$ of length $1 \geq \frac{n}{k+1}$, so this may serve as a base case. Now, for the inductive step, consider any $n' > k+1$ such that the claim in the lemma holds for all $n < n'$. Fix any $s_1, \ldots, s_{n'}$ satisfying (3) for all $i \leq n'$. By (3) for $i = n'$, there exists $\pi_{n'} \in \Pi$ with $\pi_{n'}(s_{n'}) \neq \pi_0(s_{n'})$, and for which the set $S_{\text{diff}} := \{t < n' : \pi_{n'}(s_t) \neq \pi_0(s_t)\}$ satisfies $|S_{\text{diff}}| \leq k$. Let $n = n' - 1 - |S_{\text{diff}}|$ and let $t_1, \ldots, t_n$ denote the subsequence comprised of all $t \notin S_{\text{diff}} \cup \{n'\}$. Note that since $\text{DIS}()$ is monotone in its argument, (3) remains satisfied when restricted to this subsequence: that is,

$$\forall i \leq n, \ \ s_{t_i} \in \text{DIS}\left(\left\{\pi \in \Pi : \sum_{j < i} \mathbb{1}[\pi(s_{t_j}) \neq \pi_0(s_{t_j})] \leq k\right\}\right).$$

Therefore, the inductive hypothesis implies that there exists $m \in \mathbb{N}$ with $m \geq \frac{n}{k+1}$ and a sequence $i_1 < \cdots < i_m$ in $\{t_1, \ldots, t_n\}$ such that $\forall j \in \{1, \ldots, m\}$, (4) holds. Moreover, since none of these $i_j$ are in $S_{\text{diff}} \cup \{n'\}$, we have that $\forall j \in \{1, \ldots, m\}$, $\pi_{n'}(s_{i_j}) = \pi_0(s_{i_j})$. Since $\pi_{n'}(s_{n'}) \neq \pi_0(s_{n'})$, we conclude that $s_{n'} \in \text{DIS}(\{\pi \in \Pi : \forall t < m+1, \pi(s_{i_t}) = \pi_0(s_{i_t})\})$. Thus, we may extend the above sequence by defining $i_{m+1} = n'$ while satisfying (4) for all $j \in \{1, \ldots, m+1\}$. Noting that, since $|S_{\text{diff}}| \leq k$, we have

$$m + 1 \geq \frac{n}{k+1} + 1 = \frac{n' - 1 - |S_{\text{diff}}|}{k+1} + 1 \geq \frac{n'}{k+1},$$

we have thus extended the inductive hypothesis to sequences of length $n'$, and the result follows by the principle of induction. ∎

We are now ready for the proof of Theorem 8.

**Proof of Theorem 8** We continue the notation from the proof of Theorem 6: namely, $I_t$ and $E_1$. The proof consists of three parts. The first part is to simply recall, from the proof of Theorem 6, that on the event $E_1$ of probability at least $1 - \delta$, $\pi^* \in V$ is maintained on all rounds. Second, we will argue that the sequence of queried states $Q$ satisfies the property (3) from Lemma 13 for $k = O\left(\frac{1}{\Delta^2} \log\left(\frac{|\Pi|T}{\delta}\right)\right)$ and $\pi_0 = \pi^*$. In other words, roughly speaking, we will show the set $V$ of surviving policies at any time $t \leq T$ is contained in a Hamming ball of radius $k$ centered at $\pi^*$. Finally, we will apply Lemma 13 to establish the claimed upper bound on $|Q|$.

As discussed, our second claim is effectively that the set $V$ of surviving policies at the end of each round $t$ satisfies that every $\pi \in V$ has

$$\sum_{t' \leq t} \mathbb{1}[\pi(s_{t'}) \neq \pi^*(s_{t'})]I_{t'} \leq k \tag{5}$$

where

$$k = \frac{c_1}{\Delta^2} \log\left(\frac{|\Pi|T}{\delta}\right), \tag{6}$$

for an appropriate universal constant $c_1$. Suppose the event $E_1$ holds. In particular, this means that at the conclusion of round $t$, every $\pi \in V$ satisfies

$$\sum_{t'=1}^{t} \mathbb{E}[(\mathbb{1}[\pi^*(s_{t'}) = \hat{a}_{t'}] - \mathbb{1}[\pi(s_{t'}) = \hat{a}_{t'}]) I_{t'}|s_1, \ldots, s_{t'}, \hat{a}_1, \ldots, \hat{a}_{t'-1}]$$

$$\leq \sum_{t'=1}^{t} (\mathbb{1}[\pi^*(s_{t'}) = \hat{a}_{t'}] - \mathbb{1}[\pi(s_{t'}) = \hat{a}_{t'}]) I_{t'}$$

$$+ c_0 \sqrt{\left( \sum_{t'=1}^{t} \mathbb{1}[\{\hat{a}_{t'}\} \subsetneq \{\pi^*(s_{t'}), \pi(s_{t'})\}] I_{t'} \right) \log\left(\frac{1}{\delta_t}\right)} + c_0 \log\left(\frac{1}{\delta_t}\right)$$

$$\leq \sum_{t'=1}^{t} (\mathbb{1}[\hat{\pi}_t(s_{t'}) = \hat{a}_{t'}] - \mathbb{1}[\pi(s_{t'}) = \hat{a}_{t'}]) I_{t'}$$

$$+ c_0 \sqrt{\left( \sum_{t'=1}^{t} \mathbb{1}[\{\hat{a}_{t'}\} \subsetneq \{\pi^*(s_{t'}), \pi(s_{t'})\}] I_{t'} \right) \log\left(\frac{1}{\delta_t}\right)} + c_0 \log\left(\frac{1}{\delta_t}\right).$$

Since $\pi \in V$, we further have (recalling that $c = c_0$)

$$\sum_{t'=1}^{t} (\mathbb{1}[\hat{\pi}_t(s_{t'}) = \hat{a}_{t'}] - \mathbb{1}[\pi(s_{t'}) = \hat{a}_{t'}]) I_{t'}$$

$$\leq c_0 \sqrt{\left( \sum_{t'=1}^{t} \mathbb{1}[\{\hat{a}_{t'}\} \subsetneq \{\pi(s_{t'}), \hat{\pi}_t(s_{t'})\}] I_{t'} \right) \log\left(\frac{1}{\delta_t}\right)} + c_0 \log\left(\frac{1}{\delta_t}\right).$$

Letting $c_2 = 2c_0$, together we have that

$$\sum_{t'=1}^{t} \mathbb{E}[(\mathbb{1}[\pi^*(s_{t'}) = \hat{a}_{t'}] - \mathbb{1}[\pi(s_{t'}) = \hat{a}_{t'}]) I_{t'}|s_1, \ldots, s_{t'}, \hat{a}_1, \ldots, \hat{a}_{t'-1}]$$

$$\leq c_2 \sqrt{\left( \sum_{t'=1}^{t} \mathbb{1}[\{\hat{a}_{t'}\} \subsetneq \{\hat{\pi}_t(s_{t'}), \pi^*(s_{t'}), \pi(s_{t'})\}] I_{t'} \right) \log\left(\frac{1}{\delta_t}\right)} + c_2 \log\left(\frac{1}{\delta_t}\right). \tag{7}$$

Noting that

$$\mathbb{1}[\{\hat{a}_{t'}\} \subsetneq \{\hat{\pi}_t(s_{t'}), \pi^*(s_{t'}), \pi(s_{t'})\}]$$
$$\leq \mathbb{1}[\{\hat{a}_{t'}\} \subsetneq \{\hat{\pi}_t(s_{t'}), \pi^*(s_{t'})\}] + \mathbb{1}[\{\hat{a}_{t'}\} \subsetneq \{\pi^*(s_{t'}), \pi(s_{t'})\}],$$

since both $\hat{\pi}_t$ and $\pi$ are in $V$, we have that the last expression in (7) is at most

$$c_2 \sqrt{\left( 2 \max_{\pi' \in V} \sum_{t'=1}^{t} \mathbb{1}[\{\hat{a}_{t'}\} \subsetneq \{\pi'(s_{t'}), \pi^*(s_{t'})\}] I_{t'} \right) \log\left(\frac{1}{\delta_t}\right)} + c_2 \log\left(\frac{1}{\delta_t}\right)$$

$$\leq c_2 \sqrt{\left( 2 \max_{\pi' \in V} \sum_{t'=1}^{t} \mathbb{1}[\pi'(s_{t'}) \neq \pi^*(s_{t'})] I_{t'} \right) \log\left(\frac{1}{\delta_t}\right)} + c_2 \log\left(\frac{1}{\delta_t}\right).$$

Additionally, due to the Massart noise condition (Definition 7), every $\pi \in V$ satisfies

$$\mathbb{E}[(\mathbb{1}[\pi^*(s_{t'}) = \hat{a}_{t'}] - \mathbb{1}[\pi(s_{t'}) = \hat{a}_{t'}]) I_{t'} | s_1, \ldots, s_{t'}, \hat{a}_1, \ldots, \hat{a}_{t'-1}] \geq \Delta \mathbb{1}[\pi(s_{t'}) \neq \pi^*(s_{t'})] I_{t'}.$$

Together, we have that on $E_1$,

$$\max_{\pi \in V} \sum_{t'=1}^{t} \mathbb{1}[\pi(s_{t'}) \neq \pi^*(s_{t'})] I_{t'}$$

$$\leq \frac{1}{\Delta} \left( c_2 \sqrt{\left( 2 \max_{\pi \in V} \sum_{t'=1}^{t} \mathbb{1}[\pi(s_{t'}) \neq \pi^*(s_{t'})] I_{t'} \right) \log\left(\frac{1}{\delta_t}\right)} + c_2 \log\left(\frac{1}{\delta_t}\right) \right).$$

Solving this quadratic inequality yields

$$\max_{\pi \in V} \sum_{t'=1}^{t} \mathbb{1}[\pi(s_{t'}) \neq \pi^*(s_{t'})] I_{t'} \leq \frac{c_3}{\Delta^2} \log\left(\frac{1}{\delta_t}\right)$$

for an appropriate universal constant $c_3$. Thus, we have verified that (5) holds with $k$ as in (6) (for $c_1 = c_3$).

Consider the subsequence $s_{t_j}$ of states queried by the algorithm: i.e., the states in $Q$ after all $T$ rounds. On the event $E_1$, we have established that each at the start of each round $t = t_j$, we have

$$V \subseteq \left\{ \pi \in \Pi : \sum_{j' < j} \mathbb{1}\left[ \pi(s_{t_{j'}}) \neq \pi^*(s_{t_{j'}}) \right] \leq k \right\},$$

for $k$ as in (6). Therefore, by definition of $Q$, we have that every $(s_{t_j}, a_{t_j}) \in Q$ satisfies

$$s_{t_j} \in \text{DIS}(V) \subseteq \left\{ \pi \in \Pi : \sum_{j' < j} \mathbb{1}\left[ \pi(s_{t_{j'}}) \neq \pi^*(s_{t_{j'}}) \right] \leq k \right\}.$$

Therefore, by Lemma 13, we conclude that on the event $E_1$,

$$|Q| \leq (k+1)\mathfrak{e}_{\pi^*} = O\left( \mathfrak{e}_{\pi^*} \frac{1}{\Delta^2} \log\left(\frac{|\Pi|T}{\delta}\right) \right).$$

This completes the proof. ∎

We complement the upper bound from Theorem 8 with a lower bound in Theorem 9.

The proof of Theorem 9 will be based on the following well-known lower bound for testing the bias of a Bernoulli random variable based on samples.

**Lemma 14 (Lemma 5.1 of Anthony and Bartlett, 1999)** *Let* $\Delta \in (0, 2/5)$, $\delta \in (0, 1/8e]$, *and*

$$n \leq \frac{1}{2\Delta^2} \ln\left(\frac{1}{8\delta}\right).$$

*For any function* $\hat{t} : \{0, 1\}^n \to \{0, 1\}$, *for* $b \sim \mathrm{Uniform}(\{-1, 1\})$, *for* $B_1, \ldots, B_n$ *conditionally i.i.d.* $\mathrm{Bernoulli}\left(\frac{1+b\Delta}{2}\right)$ *given* $b$,

$$\mathbb{P}\big(\hat{t}(B_1, \ldots, B_n) \neq b\big) > \delta.$$

*This further implies that, for any given* $\hat{t}$ *function, there exists a deterministic choice of* $b$ *for which* $\mathbb{P}\big(\hat{t}(B_1, \ldots, B_n) \neq b\big) > \delta$.

We now present the proof of Theorem 9.

**Proof of Theorem 9** Fix any $\pi^* \in \Pi$. We begin by showing a lower bound $\Omega(\min\{\mathfrak{e}_{\pi^*}, T\})$. While such a lower bound was already established for realizable-case reliable learning (Theorem 3), we here confirm that this lower bound remains valid even when we allow for the $\delta$ failure probability afforded to reliable learners under Massart noise. Fix any finite value $\mathfrak{e} \leq \mathfrak{e}_{\pi^*} \wedge T$, and let $s_1, \ldots, s_{\mathfrak{e}}$ be an eluder sequence centered at $\pi^*$: namely, let $\pi_1, \ldots, \pi_{\mathfrak{e}} \in \Pi$ be such that $\forall i \leq \mathfrak{e}$, $\pi_i(s_i) \neq \pi^*(s_i)$ and $\forall j < i$, $\pi_i(s_j) = \pi^*(s_j)$. Such sequences $s_i$ and $\pi_i$ are guaranteed to exist by the definition of $\mathfrak{e}_{\pi^*}$. Define an initial state distribution $P_0$ which produces $s_1$ with probability one. Define state transition distribution which, for any $i \in \{1, \ldots, \mathfrak{e} - 1\}$, satisfies that $P(s_{i+1}|s_i, a) = 1$ for every action $a \in \mathcal{A}$, and $P(s_{\mathfrak{e}}|s_{\mathfrak{e}}, a) = 1$ as well. Define query responses $P(\pi^*(s)|s) = 1$ for every $s$. This environment deterministically follows the sequence $s_1, s_2, \ldots, s_{\mathfrak{e}}$, and answers every query with the $\pi^*$ action. It therefore trivially satisfies the Massart noise condition.

Fix any learning algorithm $\mathbb{A}$. We consider two cases. First, suppose there is some $i \leq \mathfrak{e}$ such that, when $\mathbb{A}$ is run under the above environment $P$, with probability greater than $2\delta$, upon reaching state $s_i$ (for the first time, in the case of $i = \mathfrak{e}$), it does *not* query the oracle. Given the event that it does not query the oracle, if the learner has conditional probability at least $1/2$ of *not* taking action $\pi^*(s_i)$, then altogether it has probability greater than $\delta$ of not querying *and* yet not taking action $pi^*(s_i)$, and hence is not reliable under Massart noise. Otherwise, suppose that, given the event that the learner does not query the oracle upon reaching state $s_i$, it has conditional probability greater than $1/2$ of taking action $\pi^*(s_i)$. Consider an alternative environment $P'$, which is identical to $P$ except that query responses satisfy $P'(\pi_i(s)|s) = 1$ for every state $s$ ($P'$ clearly also satisfies the Massart noise condition). In particular, note that any queries among the states $s_1, \ldots, s_{i-1}$ will return identical actions under $P$ and $P'$, so that the distribution of the algorithm's behaviors upon reaching state $s_i$ will be identical under these two environments. Thus, overall, when running $\mathbb{A}$ under environment $P'$, the algorithm has probability greater than $\delta$ of not querying upon reaching $s_i$ and yet taking action $\pi^*(s_i)$, *not* the optimal action $\pi_i(s_i)$ (recalling that $\pi_i(s_i) \neq \pi^*(s_i)$), and hence is not reliable under Massart noise.

Next consider a second case: namely, suppose that, for every $i \leq \mathfrak{e}$, when $\mathbb{A}$ is run under the environment $P$, it has probability at most $2\delta$ of *not* querying the oracle upon reaching state $s_i$. Let $Q$ denote the total number of queries by the learner in the first $\mathfrak{e}$ rounds. We have $\mathbb{E}[Q] \geq (1 - 2\delta)\mathfrak{e}$. Since $Q \leq \mathfrak{e}$, we have $\mathfrak{e}\mathbb{P}(Q > (1/2)(1 - 2\delta)\mathfrak{e}) + (1/2)(1 - 2\delta)\mathfrak{e} \geq \mathbb{E}[Q] \geq (1 - 2\delta)\mathfrak{e}$, so that $\mathbb{P}(Q > (1/2)(1 - 2\delta)\mathfrak{e}) \geq (1/2)(1 - 2\delta)$, which is greater than $\delta$ if $\delta \in (0, 1/4)$. Thus, with probability greater than $\delta$, the algorithm makes at least $(1/2)(1 - 2\delta)\mathfrak{e}$ queries. Since the above analysis is valid for any $\mathfrak{e} \leq \mathfrak{e}_{\pi^*} \wedge T$, the lower bound $\Omega(\mathfrak{e}_{\pi^*} \wedge T)$ follows.

For the remaining term, if it is larger than the first term, we will witness the lower bound by constructing a $P$ as follows. Let $\pi' \in \Pi$ and $s_0 \in \mathcal{S}$ satisfy that $\pi'(s_0) \neq \pi^*(s_0)$. Such a $\pi'$ and $s_0$ must exist, since $|\Pi| \geq 2$. Define the initial state distribution of $P$ to have probability one of $s_0$, and the state transition distribution satisfies $P(s_0|s_0, a) = 1$ for every action $a$. In other words, the state sequence deterministically repeats state $s_0$. The oracle's response distribution for any state $s$ is defined as $P(\pi^*(s)|s) = \frac{1+\Delta}{2}$ and $P(\pi'(s)|s) = \frac{1-\Delta}{2}$. Note that this indeed satisfies the Massart noise condition.

Consider an alternative environment $P'$, which is identical to $P$ except that a query in any state $s$ has $P'(\pi^*(s)|s) = \frac{1-\Delta}{2}$ and $P'(\pi'(s)|s) = \frac{1+\Delta}{2}$. Again, this clearly satisfies the Massart noise condition (now with optimal policy $\pi'$). Note that $\pi^*$ is *not* an optimal policy for this environment $P'$, and rather, $\pi'$ is optimal. Any reliable learner must satisfy the reliability guarantee regardless of whether run under $P$ or $P'$, and must therefore be able to distinguish which of these cases it is in before it can reliably choose an action in state $s_0$ rather than querying.

We will establish the lower bound via a standard reduction from *hypothesis testing* (Lemma 14). We now set up the reduction from hypothesis testing. Consider running a learner $\mathbb{A}$ under an environment $\tilde{P}$ sampled uniformly at random from $\{P, P'\}$. Let $n = \left\lfloor \frac{1}{2\Delta^2} \ln\left(\frac{1}{16\delta}\right) \right\rfloor \wedge T$. For each time $t \leq n$, define a value $B_t$ which is 0 if the oracle's (hypothetical) response at that time is $\pi^*(s_0)$ and 1 if it is $\pi'(s_0)$; note that under $\tilde{P}$, these are the only possibilities. When executing the learner under environment $\tilde{P}$, if there exists some time $t \leq n$ for which it does not query the oracle, let $t$ be the earliest such time, and if its action at this time $t$ is among $\pi^*(s_0)$ or $\pi'(s_0)$, define $\hat{t}(B_1, \ldots, B_n)$ as $-1$ if its action is $\pi^*(s_0)$ and as 1 if it is $\pi'(s_0)$. If its action is not one of these, or if the learner queries at every time $t \leq n$, let $\hat{t}(B_1, \ldots, B_n) = 1$. Note that, since the learner's action is based on the oracle responses, then its behavior is indeed a (possibly randomized) function of the $B_i$ variables, so that this is a valid definition of $\hat{t}(B_1, \ldots, B_n)$. Note that if the oracle is responding with $P$ then a value $\hat{t}(B_1, \ldots, B_{\hat{k}}) = 1$ represents either an unreliable action (action different from the optimal action) or the event that it queries every time $t \leq n$ when responses are from $P$, whereas if the oracle is responding with $P'$ then a value $\hat{t}(B_1, \ldots, B_n) = -1$ implies an unreliable action.

Applying Lemma 14, with probability greater than $2\delta$ we have either that $\hat{t}(B_1, \ldots, B_n) = 1$ while the environment is $P$ or that $\hat{t}(B_1, \ldots, B_n) = -1$ while the environment is $P'$. In particular, by the law of total probability, this means either (1) the probability of an unreliable action under $P'$ is greater than $2\delta$ (so that the learner is not reliable under Massart noise), or (2) there is an event $E$ of probability greater than $2\delta$ under $P$ that the algorithm either makes an unreliable action (for $P$) or queries in all of the first $n$ rounds. In this second case, if we suppose the learner is reliable under Massart noise, it has an event $E'$ of probability at least $1 - \delta$ on which it never makes an unreliable action on a non-querying round. In particular, this implies $E \cap E'$ has probability greater than $\delta$, and on the event $E \cap E'$, it must be that the algorithm queries in all of the first $n$ rounds: that is, with probability greater than $\delta$, it makes a number of queries at least $n$. This completes the proof of the lower bound. ∎

## 6. The Mixed Margin Condition

This section presents the proof of Theorems 11 and 12, establishing upper and lower bounds on the optimal query complexity under the mixed-margin condition. A main ingredient in the proof of Theorem 11 is an analogue of the *Bernstein class condition* (Tsybakov, 2004; Bousquet, Boucheron,

and Lugosi, 2004; Bartlett, Jordan, and McAuliffe, 2006), appropriately formulated to relate to Definition 10.

**Definition 15** *We say $P$ satisfies the mixed Bernstein class condition with parameters $(C', \alpha) \in [1, \infty) \times [0, 1)$ if, for every mixed optimal trajectory $(\mathbf{s}_1, \mathbf{a}_1, \ldots, \mathbf{s}_T, \mathbf{s}_T)$, for every $t \leq T$ and $\delta' \in (0, 1)$, with probability at least $1 - \delta'$, every $\pi \in \Pi$ satisfies*

$$\frac{1}{t} \sum_{t'=1}^{t} \mathbb{1}[\pi(\mathbf{s}_{t'}) \neq \pi^*(\mathbf{s}_{t'})] \leq C' \left( \frac{1}{t} \sum_{t'=1}^{t} (P(\pi^*(\mathbf{s}_{t'})|\mathbf{s}_{t'}) - P(\pi(\mathbf{s}_{t'})|\mathbf{s}_{t'})) \right)^{\alpha} + \frac{1}{t} \log\left( \frac{1}{\delta'} \right).$$

Analogously to the Bernstein class condition in classification, the margin condition implies the Bernstein class condition. Specifically, we have the following lemma.

**Lemma 16** *For any $P$ satisfying the mixed margin condition (Definition 10) with parameters $(C, \alpha) \in [1, \infty) \times (0, 1)$, for every mixed optimal trajectory $(\mathbf{s}_1, \mathbf{a}_1, \ldots, \mathbf{s}_T, \mathbf{a}_T)$, for every $t \leq T$, $\delta' \in (0, 1)$, and $\pi \in \Pi$, with probability at least $1 - \delta'$,*

$$\frac{1}{t} \sum_{t'=1}^{t} \mathbb{1}[\pi(\mathbf{s}_{t'}) \neq \pi^*(\mathbf{s}_{t'})] \leq C' \left( \frac{1}{t} \sum_{t'=1}^{t} (P(\pi^*(\mathbf{s}_{t'})|\mathbf{s}_{t'}) - P(\pi(\mathbf{s}_{t'})|\mathbf{s}_{t'})) \right)^{\alpha} + \frac{1}{t} \log\left( \frac{1}{\delta'} \right),$$

*where $C' = C^{1-\alpha}(1 - \alpha)^{\alpha-1}\alpha^{-\alpha}$.*

**Proof** Suppose $P$ satisfies Definition 10. Consider a mixed optimal trajectory $(\mathbf{s}_1, \mathbf{a}_1, \ldots, \mathbf{s}_T, \mathbf{a}_T)$. Let $t \leq T$, $\delta' \in (0, 1)$, and $\pi \in \Pi$. By Definition 10, with probability at least $1 - \delta'$, for every $\tau > 0$,

$$\frac{1}{t} \sum_{t'=1}^{t} \mathbb{1}\left[ P(\pi^*(\mathbf{s}_{t'})|\mathbf{s}_{t'}) - \max_{a \neq \pi^*(\mathbf{s}_{t'})} P(a|\mathbf{s}_{t'}) \leq \tau \right] \leq C\tau^{\frac{\alpha}{1-\alpha}} + \frac{1}{t} \log\left( \frac{1}{\delta'} \right). \tag{8}$$

Suppose this event holds.

Note that for any $\tau > 0$,

$$\frac{1}{t} \sum_{t'=1}^{t} (P(\pi^*(\mathbf{s}_{t'})|\mathbf{s}_{t'}) - P(\pi(\mathbf{s}_{t'})|\mathbf{s}_{t'}))$$

$$\geq \tau \frac{1}{t} \sum_{t'=1}^{t} \mathbb{1}[\pi(\mathbf{s}_{t'}) \neq \pi^*(\mathbf{s}_{t'})] \mathbb{1}[P(\pi^*(\mathbf{s}_{t'})|\mathbf{s}_{t'}) - P(\pi(\mathbf{s}_{t'})|\mathbf{s}_{t'}) \geq \tau]$$

$$\geq \tau \left( \frac{1}{t} \sum_{t'=1}^{t} \mathbb{1}\left[ P(\pi^*(\mathbf{s}_{t'})|\mathbf{s}_{t'}) - \max_{a \neq \pi^*(\mathbf{s}_{t'})} P(a|\mathbf{s}_{t'}) \geq \tau \right] \right) - \tau \left( \frac{1}{t} \sum_{t'=1}^{t} \mathbb{1}[\pi(\mathbf{s}_{t'}) = \pi^*(\mathbf{s}_{t'})] \right)$$

$$\geq \tau \left( 1 - C\tau^{\frac{\alpha}{1-\alpha}} - \frac{1}{t} \log\left( \frac{1}{\delta'} \right) \right) - \tau \left( \frac{1}{t} \sum_{t'=1}^{t} \mathbb{1}[\pi(\mathbf{s}_{t'}) = \pi^*(\mathbf{s}_{t'})] \right)$$

$$= \tau \left( \frac{1}{t} \sum_{t'=1}^{t} \mathbb{1}[\pi(\mathbf{s}_{t'}) \neq \pi^*(\mathbf{s}_{t'})] \right) - C\tau^{1+\frac{\alpha}{1-\alpha}} - \frac{\tau}{t} \log\left( \frac{1}{\delta'} \right).$$

Denoting by

$$\mathbf{A} = \frac{1}{t} \sum_{t'=1}^{t} \mathbb{1}[\pi(\mathbf{s}_{t'}) \neq \pi^*(\mathbf{s}_{t'})]$$

and setting

$$\tau = \left( \frac{(1-\alpha)}{C} \left( \mathbf{A} - \frac{1}{t} \log\left(\frac{1}{\delta'}\right) \right) \right)^{\frac{1-\alpha}{\alpha}}$$

gives

$$\frac{1}{t} \sum_{t'=1}^{t} \left( P(\pi^*(\mathbf{s}_{t'})|\mathbf{s}_{t'}) - P(\pi(\mathbf{s}_{t'})|\mathbf{s}_{t'}) \right)$$

$$\geq \left( \frac{(1-\alpha)}{C} \left( \mathbf{A} - \frac{1}{t} \log\left(\frac{1}{\delta'}\right) \right) \right)^{\frac{1-\alpha}{\alpha}} \mathbf{A} - C \left( \frac{(1-\alpha)}{C} \left( \mathbf{A} - \frac{1}{t} \log\left(\frac{1}{\delta'}\right) \right) \right)^{\frac{1}{\alpha}}$$

$$- \left( \frac{(1-\alpha)}{C} \left( \mathbf{A} - \frac{1}{t} \log\left(\frac{1}{\delta'}\right) \right) \right)^{\frac{1-\alpha}{\alpha}} \frac{1}{t} \log\left(\frac{1}{\delta'}\right)$$

$$= C^{\frac{\alpha-1}{\alpha}} (1-\alpha)^{\frac{1}{\alpha}} \left( \frac{\alpha}{1-\alpha} \right) \left( \mathbf{A} - \frac{1}{t} \log\left(\frac{1}{\delta'}\right) \right)^{\frac{1}{\alpha}}.$$

Solving for $\mathbf{A}$ yields

$$\mathbf{A} \leq C^{1-\alpha} (1-\alpha)^{\alpha-1} \alpha^{-\alpha} \left( \frac{1}{t} \sum_{t'=1}^{t} \left( P(\pi^*(\mathbf{s}_{t'})|\mathbf{s}_{t'}) - P(\pi(\mathbf{s}_{t'})|\mathbf{s}_{t'}) \right) \right)^{\alpha} + \frac{1}{t} \log\left(\frac{1}{\delta'}\right).$$

■

We are now ready for the proof of Theorem 11. In particular, Theorem 11 follows immediately from a combination of Lemma 16 and the following result.

**Lemma 17** *For any $P$ satisfying the mixed Bernstein class condition (Definition 15) with parameters $(C', \alpha)$, on the event $E_1$ (of probability at least $1-\delta$) from the proof of Theorem 6, the algorithm ReliableApprentice is reliable, and on an additional event $E_2$ of probability at least $1 - \delta$, makes a number of oracle queries at most*

$$O \left( \mathfrak{e}_{\pi^*} \cdot (C')^{\frac{2}{2-\alpha}} T^{\frac{2-2\alpha}{2-\alpha}} \left( \log\left( \frac{|\Pi|T}{\delta} \right) \right)^{\frac{\alpha}{2-\alpha}} \right).$$

**Proof of Lemma 17** The proof follows analogously to the proof of Theorem 8. We will again have three main parts: (1) arguing that $\pi^* \in V$ is preserved as an invariant, (2) arguing that on round $t$, $V$ is contained in a Hamming ball of some radius $k_t$, and (3) bounding the number of queries based on this fact via Lemma 13.

As before, the first property will hold via martingale concentration inequalities. Let $I_t \in \{0, 1\}$ be 1 iff $s_t \in \text{DIS}(V)$ on round $t$. Let $E_1$ be as in the proof of Theorem 8. Note that this portion of the proof of Theorem 8, establishing that $E_1$ holds with probability at least $1 - \frac{\delta}{2}$, and that $\pi^* \in V$ is

maintained as an invariant on the event $E_1$, remains valid under the mixed Bernstein class condition as well. In particular, from the definition of the algorithm, it follows that ReliableApprentice is reliable.

Next we establish the second claim, again via an argument similar to that in the proof of Theorem 8, but in this case requiring some modifications. Let $V_t$ denote the set $V$ at the end of round $t$. Suppose the event $E_1$ holds. As argued above, this also implies $\pi^* \in V_t$ on every round. As in the proof of Theorem 8, at the conclusion of round $t$, every $\pi \in V_t$ satisfies

$$\sum_{t'=1}^{t} \mathbb{E}[(\mathbb{1}[\pi^*(s_{t'}) = \hat{a}_{t'}] - \mathbb{1}[\pi(s_{t'}) = \hat{a}_{t'}]) I_{t'} | s_1, \ldots, s_{t'}, \hat{a}_1, \ldots, \hat{a}_{t'-1}]$$

$$\leq c_2 \sqrt{\left( \max_{\pi' \in V_t} \sum_{t'=1}^{t} \mathbb{1}[\pi'(s_{t'}) \neq \pi^*(s_{t'})] I_{t'} \right) \log\left( \frac{1}{\delta_t} \right) + c_2 \log\left( \frac{1}{\delta_t} \right)}$$

$$= c_2 \sqrt{\left( \max_{\pi' \in V_t} \sum_{t'=1}^{t} \mathbb{1}[\pi'(s_{t'}) \neq \pi^*(s_{t'})] \right) \log\left( \frac{1}{\delta_t} \right) + c_2 \log\left( \frac{1}{\delta_t} \right)},$$

for a universal constant $c_2$, where the final equality holds because $\pi' \in V_t$ implies that $\pi'(s_{t'}) = \pi^*(s_{t'})$ for any $t' \leq t$ with $I_{t'} = 0$ (since $V_t$ is non-increasing in the rounds $t$). Additionally, since $\hat{a}_{t'} \sim P(\cdot|s_{t'})$ whenever $I_{t'} = 1$, we have

$$\sum_{t'=1}^{t} \mathbb{E}[(\mathbb{1}[\pi^*(s_{t'}) = \hat{a}_{t'}] - \mathbb{1}[\pi(s_{t'}) = \hat{a}_{t'}]) I_{t'} | s_1, \ldots, s_{t'}, \hat{a}_1, \ldots, \hat{a}_{t'-1}]$$

$$= \sum_{t'=1}^{t} (P(\pi^*(s_{t'})|s_{t'}) - P(\pi(s_{t'})|s_{t'})) I_{t'} = \sum_{t'=1}^{t} (P(\pi^*(s_{t'})|s_{t'}) - P(\pi(s_{t'})|s_{t'})),$$

where the final equality is again due to the fact that $\pi \in V_t$, which implies $I_{t'} = 0 \implies \pi^*(s_{t'}) = \pi(s_{t'}) \implies P(\pi^*(s_{t'})|s_{t'}) - P(\pi(s_{t'})|s_{t'}) = 0$. Thus, on the event $E_1$, we have

$$\max_{\pi \in V_t} \sum_{t'=1}^{t} (P(\pi^*(s_{t'})|s_{t'}) - P(\pi(s_{t'})|s_{t'}))$$

$$\leq c_2 \sqrt{\left( \max_{\pi \in V_t} \sum_{t'=1}^{t} \mathbb{1}[\pi(s_{t'}) \neq \pi^*(s_{t'})] \right) \log\left( \frac{1}{\delta_t} \right) + c_2 \log\left( \frac{1}{\delta_t} \right)}. \tag{9}$$

Additionally, note that since the algorithm is reliable, on the event $E_1$, $s_1, \ldots, s_T$ follows a mixed optimal trajectory. Therefore, Definition 15 and the union bound imply that on an event $E_2$ of probability at least $1 - \sum_{t \leq T} \delta_t \geq 1 - \frac{\delta}{2}$, every $t \leq T$ and $\pi \in \Pi$ satisfies

$$\frac{1}{t} \sum_{t'=1}^{t} \mathbb{1}[\pi(s_{t'}) \neq \pi^*(s_{t'})] \leq C' \left( \frac{1}{t} \sum_{t'=1}^{t} (P(\pi^*(s_{t'})|s_{t'}) - P(\pi(s_{t'})|s_{t'})) \right)^{\alpha} + \frac{1}{t} \log\left( \frac{1}{\delta_t} \right).$$

Plugging this into (9) and simplifying gives that, on the event $E_1 \cap E_2$,

$$\max_{\pi \in V_t} \frac{1}{t} \sum_{t'=1}^{t} \left( P(\pi^*(s_{t'})|s_{t'}) - P(\pi(s_{t'})|s_{t'}) \right)$$

$$\leq 2c_2 \sqrt{C' \left( \max_{\pi \in V_t} \frac{1}{t} \sum_{t'=1}^{t} \left( P(\pi^*(s_{t'})|s_{t'}) - P(\pi(s_{t'})|s_{t'}) \right) \right)^{\alpha} \frac{1}{t} \log\left(\frac{1}{\delta_t}\right) + \frac{3c_2}{t} \log\left(\frac{1}{\delta_t}\right)}$$

$$\leq \max\left\{ 4c_2 \sqrt{C' \left( \max_{\pi \in V_t} \frac{1}{t} \sum_{t'=1}^{t} \left( P(\pi^*(s_{t'})|s_{t'}) - P(\pi(s_{t'})|s_{t'}) \right) \right)^{\alpha} \frac{1}{t} \log\left(\frac{1}{\delta_t}\right)}, \frac{6c_2}{t} \log\left(\frac{1}{\delta_t}\right) \right\}.$$

In the case the first term in the $\max$ is larger, we may simplify the inequality (i.e., $x \leq a\sqrt{bx^{\alpha}} \implies x \leq (a^2 b)^{\frac{1}{2-\alpha}}$), so that altogether we have

$$\max_{\pi \in V_t} \frac{1}{t} \sum_{t'=1}^{t} \left( P(\pi^*(s_{t'})|s_{t'}) - P(\pi(s_{t'})|s_{t'}) \right)$$

$$\leq \max\left\{ \left( 16c_2^2 C' \frac{1}{t} \log\left(\frac{1}{\delta_t}\right) \right)^{\frac{1}{2-\alpha}}, \frac{6c_2}{t} \log\left(\frac{1}{\delta_t}\right) \right\}$$

$$\leq \left( 16c_2^2 C' \frac{1}{t} \log\left(\frac{1}{\delta_t}\right) \right)^{\frac{1}{2-\alpha}} + \frac{6c_2}{t} \log\left(\frac{1}{\delta_t}\right).$$

In particular, on the event $E_2$, this further implies

$$\max_{\pi \in V_t} \frac{1}{t} \sum_{t'=1}^{t} \mathbb{1}[\pi(s_{t'}) \neq \pi^*(s_{t'})]$$

$$\leq C' \left( \left( 16c_2^2 C' \frac{1}{t} \log\left(\frac{1}{\delta_t}\right) \right)^{\frac{1}{2-\alpha}} + \frac{6c_2}{t} \log\left(\frac{1}{\delta_t}\right) \right)^{\alpha} + \frac{1}{t} \log\left(\frac{1}{\delta_t}\right)$$

$$\leq c_3 (C')^{\frac{2}{2-\alpha}} \left( \frac{c_4}{t} \log\left(\frac{1}{\delta_t}\right) \right)^{\frac{\alpha}{2-\alpha}} \qquad (10)$$

for appropriate universal constants $c_3, c_4$.

The utility of (10) is that it enables us to bound the number of queries in terms of $\mathfrak{e}_{\pi^*}(\Pi)$ via Lemma 13, as follows. The sequence of query times $t_1 < t_2 < \cdots$ (i.e., those $t$ with $I_t = 1$) satisfy $x_{t_i} \in \mathrm{DIS}(V_{t_i-1})$. Moreover, by (10), every $\pi \in V_{t_i-1}$ satisfies

$$\sum_{j=1}^{i-1} \mathbb{1}\left[ \pi(s_{t_j}) \neq \pi^*(s_{t_j}) \right] \leq \sum_{t'=1}^{t_i-1} \mathbb{1}[\pi(s_{t'}) \neq \pi^*(s_{t'})]$$

$$\leq c_3 (C')^{\frac{2}{2-\alpha}} (t_i - 1)^{\frac{2-2\alpha}{2-\alpha}} \left( c_4 \log\left(\frac{1}{\delta_{t_i-1}}\right) \right)^{\frac{\alpha}{2-\alpha}}$$

$$\leq c_3 (C')^{\frac{2}{2-\alpha}} T^{\frac{2-2\alpha}{2-\alpha}} \left( c_4 \log\left(\frac{1}{\delta_T}\right) \right)^{\frac{\alpha}{2-\alpha}} =: k.$$

Thus, by monotonicity of $\mathrm{DIS}(\cdot)$, each $t_i$ satisfies

$$s_{t_i} \in \mathrm{DIS}\left(\left\{\pi \in \Pi : \sum_{j<i} \mathbb{1}\big[\pi(s_{t_j}) \neq \pi^*(s_{t_j})\big] \leq k\right\}\right),$$

and therefore Lemma 13 implies the total number $n$ of query times $t_i$ is at most

$$(k+1)\mathfrak{e}_{\pi^*} = O\left(\mathfrak{e}_{\pi^*} \cdot (C')^{\frac{2}{2-\alpha}} T^{\frac{2-2\alpha}{2-\alpha}} \left(\log\left(\frac{1}{\delta_T}\right)\right)^{\frac{\alpha}{2-\alpha}}\right).$$

This completes the proof. $\blacksquare$

Next we turn to establishing the lower bound stated in Theorem 12 under the mixed margin condition. The proof will again be based on the lower bound in Lemma 14 for *hypothesis testing*.

**Proof of Theorem 12** A lower bound $\Omega(\min\{\mathfrak{e}_{\pi^*}, T\})$ follows identically to the proof of Theorem 9 (noting that the realizable case always satisfies the mixed margin condition). For the remaining term, if it is larger than the first term, we will witness the lower bound by constructing a $P$ whose state transitions are supported on the two states $s_0, s_1$, as follows. Suppose $T \geq 64 \cdot \max\left\{(5/2)^{\frac{2-\alpha}{1-\alpha}}, 16^{\frac{2-\alpha}{\alpha}}\right\}$. For simplicity we will establish the result specifically for $C = 64$, which immediately implies it also for any larger $C$ (since $P$ satisfying Definition 10 with $C = 64$ also satisfies it for any larger $C$). Let $P_S$ denote a fixed distribution on the state space $\mathcal{S}$, supported on $\{s_0, s_1\}$, with $P_S(\{s_1\}) = \frac{C}{8}(C/T)^{\frac{\alpha}{2-\alpha}}$ and $P_S(\{s_0\}) = 1 - P_S(\{s_1\})$. Define the initial state distribution in $P$ to be $P_S$, and for any action $a$ from any state $s$, the transition distribution is also $P_S$: that is, regardless of actions, the state sequence $\mathbf{s}_1, \mathbf{s}_2, \ldots, \mathbf{s}_T$ is i.i.d. $P_S$.

Now for the oracle's response distribution, a query in state $s_1$ returns $\pi^*(s_1)$ with probability $\frac{1}{2} + \frac{1}{2}(C/T)^{\frac{1-\alpha}{2-\alpha}}$, and otherwise returns $\pi_1(s_1)$. A query from state $s_0$ deterministically returns $\pi^*(s_0)$. Since the state sequence is supported only on $\{s_0, s_1\}$, this completely specifies the transition distribution and query response distribution, and hence completely specifies $P$.

It remains to establish (1) that $P$ satisfies the mixed margin condition, and (2) that the claimed lower bound is satisfied by any reliable learner. For (1), note that *every* mixed-optimal trajectory $(\mathbf{s}_1, \mathbf{a}_1, \ldots, \mathbf{s}_T, \mathbf{a}_T)$ has $\mathbf{s}_1, \ldots, \mathbf{s}_T$ i.i.d. $P_S$. Additionally, note that we have $P(\pi^*(s_0)|s_0) - \max_{a \neq \pi^*(s_0)} P(a|s_0) = 1$, whereas

$$P(\pi^*(s_1)|s_1) - \max_{a \neq \pi^*(s_1)} P(a|s_1) = P(\pi^*(s_1)|s_1) - P(\pi_1(s_1)|s_1) = (C/T)^{\frac{1-\alpha}{2-\alpha}}.$$

Moreover, for any $t \leq T$ and $\delta \in (0, 1)$, by a Chernoff bound, with probability at least $1 - \delta$,

$$\frac{1}{t}\sum_{t'=1}^{t} \mathbb{1}[\mathbf{s}_{t'} = s_1] \leq 2eP_S(\{s_1\}) + \frac{1}{t}\log_2\left(\frac{1}{\delta}\right) \leq C\left(\frac{C}{T}\right)^{\frac{\alpha}{2-\alpha}} + \frac{1}{t}\log_2\left(\frac{1}{\delta}\right).$$

In particular, for any $\tau < (C/T)^{\frac{1-\alpha}{2-\alpha}}$,

$$\frac{1}{t}\sum_{t'=1}^{t} \mathbb{1}\left[P(\pi^*(\mathbf{s}_{t'})|\mathbf{s}_{t'}) - \max_{a \neq \pi^*(\mathbf{s}_{t'})} P(a|\mathbf{s}_{t'}) \leq \tau\right] = 0,$$

whereas, on the above event of probability at least $1 - \delta$, any $1 > \tau \geq (C/T)^{\frac{1-\alpha}{2-\alpha}}$ satisfies

$$\frac{1}{t} \sum_{t'=1}^{t} \mathbb{1}\left[ P(\pi^*(\mathbf{s}_{t'})|\mathbf{s}_{t'}) - \max_{a \neq \pi^*(\mathbf{s}_{t'})} P(a|\mathbf{s}_{t'}) \leq \tau \right]$$

$$= \frac{1}{t} \sum_{t'=1}^{t} \mathbb{1}[\mathbf{s}_{t'} = s_1] \leq C\left(\frac{C}{T}\right)^{\frac{\alpha}{2-\alpha}} + \frac{1}{t} \log_2\left(\frac{1}{\delta}\right) \leq C\tau^{\frac{\alpha}{1-\alpha}} + \frac{1}{t} \log\left(\frac{1}{\delta}\right).$$

Thus, $P$ indeed satisfies the mixed margin condition with parameters $(C, \alpha)$.

It remains only to establish (2): that the claimed lower bound is satisfied by any reliable learner, under this $P$. Consider an alternative environment $P_1$, which is identical to $P$ except that a query in state $s_1$ returns $\pi_1(s_1)$ with probability $\frac{1}{2} + \frac{1}{2}(C/T)^{\frac{1-\alpha}{2-\alpha}}$, and otherwise returns $\pi^*(s_1)$. Note that $\pi^*$ is *not* an optimal policy for this environment $P_1$, and rather, $\pi_1$ is optimal. Further note that, by symmetry, $P_1$ also satisfies the mixed margin condition with parameters $(C, \alpha)$. Any reliable learner must satisfy the reliability guarantee regardless of whether run under $P$ or $P_1$, and must therefore be able to distinguish which of these cases it is in before it can reliably choose an action in state $s_1$ rather than querying. Also note that the state trajectory $\mathbf{s}_1, \ldots, \mathbf{s}_T$ is independent of the actions of the learner and the responses to queries. In particular, let $\hat{k}$ denote the number of times $t$ with $\mathbf{s}_t = s_1$. By Chernoff bounds,

$$\mathbb{P}\left( \hat{k} < (1/2) P_S(\{s_1\}) T \right) \leq \exp\{ -P_S(\{s_1\}) T/8 \} \leq e^{-1},$$

and

$$\mathbb{P}\left( \hat{k} \geq 2 P_S(\{s_1\}) T \right) \leq \exp\{ -P_S(\{s_1\}) T/3 \} \leq e^{-1},$$

so that a union bound implies that, on an event $E$ of probability at least $1 - 2e^{-1} > 0$,

$$\frac{1}{16} C^{\frac{2}{2-\alpha}} T^{\frac{2-2\alpha}{2-\alpha}} = (1/2) P_S(\{s_1\}) T \leq \hat{k} \leq 2 P_S(\{s_1\}) T = \frac{1}{4} C^{\frac{2}{2-\alpha}} T^{\frac{2-2\alpha}{2-\alpha}}.$$

Denote by $t_1 < \cdots < t_{\hat{k}}$ the sequence of times $t$ with $\mathbf{s}_t = s_1$.

We will establish the lower bound via a standard reduction from *hypothesis testing* (following closely to a standard line of reasoning from statistical learning under Tsybakov noise; see e.g., Massart and Nédélec (2006); Hanneke (2014)). Specifically, we apply Lemma 14 under the conditional distribution given $\Sigma := (\hat{k}, t_1, \ldots, t_{\hat{k}})$, on the event $E$. In particular, for

$$\Delta = (C/T)^{\frac{1-\alpha}{2-\alpha}},$$

on the event $E$ we have

$$\hat{k} \leq \frac{1}{4} C^{\frac{2}{2-\alpha}} T^{\frac{2-2\alpha}{2-\alpha}} = 2^{14} (T/C)^{\frac{2-2\alpha}{2-\alpha}}$$

where the last inequality comes from our choice of $C = 64$.

We now set up the reduction from hypothesis testing. For each time $t_i$ among $t_1, \ldots, t_{\hat{k}}$, define a value $B_i$ which is 0 if the oracle's (hypothetical) response at that time is $\pi^*(s_1)$ and 1 if it is $\pi_1(s_1)$; note that under $P$ and $P_1$, these are the only possibilities. Consider choosing the environment $\tilde{P}$ to run the learner under uniformly at random from $\{P, P_1\}$. Consider running any given learner under this environment $\tilde{P}$. If there exists some time among $t_1, \ldots, t_{\hat{k}}$ for which it does not query

the oracle, let $t$ be the earliest such time, and if its action at this time $t$ is among $\pi^*(s_1)$ or $\pi_1(s_1)$, define $\hat{t}(B_1, \ldots, B_{\hat{k}})$ as 0 if its action is $\pi^*(s_1)$ and as 1 if it is $\pi_1(s_1)$. If its action is not one of these, or if the learner queries every $t_i$, let $\hat{t}(B_1, \ldots, B_{\hat{k}}) = 1$. Note that, since the learner's action is based on the state trajectory and oracle responses, then conditional on the state trajectory the action is indeed a (possibly randomized) function of the $B_i$ variables, so that this is a valid definition of $\hat{t}(B_1, \ldots, B_{\hat{k}})$. Note that if the oracle is responding with $P$ then a value $\hat{t}(B_1, \ldots, B_{\hat{k}}) = 1$ represents either an unreliable action (action different from the optimal action) or the event that it queries every time $t_i$ when responses are from $P$, whereas if the oracle is responding with $P_1$ then a value $\hat{t}(B_1, \ldots, B_{\hat{k}}) = 0$ implies an unreliable action. Applying Lemma 14 under the conditional distribution given the event $E$, it is the case that with probability greater than $\frac{1}{8} \exp\{-2^{15}\}$ we have either $\hat{t}(B_1, \ldots, B_{\hat{k}}) = 1$ while the environment is $P$ or that $\hat{t}(B_1, \ldots, B_{\hat{k}}) = 0$ while the environment is $P_1$. All of this occurs on the event $E$, so that overall the learner has conditional probability (given $E$) at least $(1 - 2e^{-1})\frac{1}{8} \exp\{-2^{15}\}$ of satisfying either (1) $\tilde{P} = P$, and either the learner queries at every time $t_1, \ldots, t_{\hat{k}}$ or makes an unreliable action (for $P$) at its first time $t_i$ at which it does not query, or (2) $\tilde{P} = P_1$, and it does not query every time $t_i$ and makes an unreliable (for $P_1$) action at the first time $t_i$ it does not query. In particular, by the law of total probability, this further implies that for any such learner, either it has a conditional probability (given $E$) at least $\frac{1}{8} \exp\{-2^{15}\}$ of being unreliable under $P_1$, or it has conditional probability (given $E$) at least $\frac{1}{8} \exp\{-2^{15}\}$ of either being unreliable or querying every time $t_i$ under $P$. Therefore, since $E$ has probability at least $1 - 2e^{-1}$, for $\delta < (1 - 2e^{-1})\frac{1}{16} \exp\{-2^{15}\}$, any reliable learner must have conditional probability (given $E$) at least $\frac{1}{16} \exp\{-2^{15}\}$ of querying every time $t_i$ under $P$. Since $E$ has probability at least $1 - 2e^{-1}$, and $\hat{k} \geq \frac{1}{16} C^{\frac{2}{2-\alpha}} T^{\frac{2-2\alpha}{2-\alpha}}$ on the event $E$, we conclude that any reliable learner must have probability at least $(1 - 2e^{-1})\frac{1}{16} \exp\{-2^{15}\}$ of querying at least $\frac{1}{16} C^{\frac{2}{2-\alpha}} T^{\frac{2-2\alpha}{2-\alpha}}$ times under $P$. ∎

We remark that there is a gap between the upper and lower bounds for the mixed margin condition. This situation is analogous to one that persisted in the active learning literature for a number of years (Hanneke, 2014), and was resolved by the work of Hanneke and Yang (2015), establishing that the lower bound is sharp. We conjecture that a similar resolution is possible for reliable active apprenticeship learning under the mixed margin condition (though this may require a different algorithm).

**Conjecture 18** *We conjecture the optimal query complexity of reliable active apprenticeship learning under the mixed margin condition always matches the lower bound in Theorem 12 up to log factors: that is, an upper bound holds, matching the lower bound up to a factor* $\mathrm{polylog}(T, |\Pi|, 1/\delta)$.

## 7. Conclusions and Future Directions

We have proposed a new learning setting, termed *reliable active apprenticeship learning*, and established near-optimal query complexity guarantees, under increasingly-general noise models: realizable, Massart, and mixed-margin. A number of important future directions remain open. Perhaps the clearest question is the extension to the *agnostic* setting, where there are *no assumptions* on the oracle. A primary challenge in such a setting is even to formulate what kind of reliability requirement is *possible*. The main difficulty in formulating this setting is that the notion of *optimal policy* $\pi^*$ depends heavily on the specific trajectory followed by the learner (in contrast to preferential

noisy oracles, where $\pi^*$ is defined to be optimal in a point-wise fashion). We leave as an important future direction the formulation of such a definition of reliability applicable (and achievable) without any assumptions on the environment $P$, including the oracle's responses.

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
