# OpenReview forum: "Reliable Active Apprenticeship Learning"
_algorithmiclearningtheory.org/ALT/2025/Conference — ALT 2025_

### Official Review · Reviewer_GEdn · 2024-10-31

**Rating:** 6
**Confidence:** 3

**Review:**

This paper introduces a learning model in which an apprentice aims to learn an optimal strategy having access to an oracle/expert which (perhaps approximately) gives examples of the optimal policy. As a major constraint, the apprentice must be reliable, that is always either follow the recommendation of the expert, or follow an action that is optimal. The goal is to ensure reliability while making as few calls to the expert as possible, knowing that the optimal policy lies in a policy class.

When the oracle is noiseless (always outputs the optimal action for the current state), they show that the optimal query complexity is exactly characterized by the eluder dimension of the policy class. In the presence of noise, they consider two settings: a Massart noise condition (probability gap between optimal action and other actions) and a Tsybakov noise condition. In both noise models, they provide query complexities depending on the eluder dimension of the policy class, and noise parameters. The upper and lower query complexity bounds do not perfectly match but share very similar form (in terms of dependencies in all parameters)


The paper is well written, introduces a clean model for reliable active learning and provides relatively sharp query complexities under the noise model considered (exactly sharp for noiseless). My main comment is that the model does not seem significantly richer/bring significantly novel insights compared to previous models in active/imitation learning both in terms of the results and the analysis. The result in the noiseless case (Thm 3) is almost a tautology: because the apprentice cannot afford to make mistakes, acting on its own only when all potential optimal policies agree is the only reasonable strategy and unsurprisingly gives the optimal (worst-case) query complexity, which is exactly the definition of the eluder dimension. The analysis for the other noise conditions does not seem particularly specific to the apprenticeship model either (although the noise definition needs to be modified for the Tsybakov noise to account for all possible reliable trajectories).


Some specific comments:

- p7 First sentence, "As the" -> "The"?

- p18 proof of Lemma 16. The assupmtion from Definition 10 holds for a fixed value of $\tau$ but in the rest of the proof, it is applied for a value that depends on $A$ which is itself random. There seems to be a mistake here.

**Paper Award:**

No

---

### Official Review · Reviewer_P1mS · 2024-11-09
**Decision making in MDP with oracle from experts**

**Rating:** 7
**Confidence:** 3

**Review:**

$\textbf{Model Definition.}$ The model proposed by the paper is as follows. A learning algorithm is required to behave optimally within a markov decision process (MDP). At every step, the learning algorithm can either make a decision on its own, or query an oracle that provides information of the optimal action associated with the current state of the MDP. While the algorithm is required to make no mistakes, the goal is to minimize the total number of oracles used within T time steps.

$\textbf{Results.}$ When the oracle is noiseless, the authors show that the query complexity of the learning model is sharply characterized by the eluder dimension of the class of possible policies of the MDP.
When the oracle is noisy but outputs the optimal action with probability larger than those of the other actions, the authors show that the agent could still succeed in making 0 mistakes with high probability, but the query complexity will depend on the specific structure of the noise added.
In particular, the authors give upper and lower bounds for both Massart noise (when the probability of the optimal action exceeds by those of the others by a fixed margin $\Delta$), and Tsybakov noise (when the optimality gaps for certain states can be arbitrarily small as long as these states rarely occur in possible trajectories of the learner). The bounds do not match with each other tightly but have the same linear dependency on the eluder dimension of the policy class. Overall, I think the work is a good contribution and leave many interesting future directions.

$\textbf{Strengths.}$ The model is natural, well motivated, and has close connections to other parts of learning theory including reliable learning, active learning and imitation learning. The algorithm is a natural implementation of the principle of disagreement-based learning, and is shown to be optimal in the realizable case.

$\textbf{Weakness.}$ There are gaps in the query complexity bounds for both the Massart noise and Tsybakov noise model. For Massart noise, the upper bound is given by the product of the eluder dimension and a statistical estimation term (related to the failure probability $\delta$ and the optimality gap $\Delta$) while the lower bound is the sum of the two. Similar issues also appear in the bounds for Tsybakov noise. The authors note that similar issues persisted in the active learning literature for a number of years, suggesting that resolution to the gap may require novel and non-trivial technical ideas.

$\textbf{Question.}$ The authors conjecture that the lower bound for Tsybakov noise is more likely to be optimal. Do the authors believe that the lower bound for Massart noise is optimal?
Suppose the policy class is completely unstructured. Doesn’t that mean the agent is forced to learn the best action for each state from the oracle separately, which requires $|S| / \Delta^2$ many queries. It seems to me this is suggesting a stronger lower bound may exist?

**Paper Award:**

No

---

### Official Review · Reviewer_Szof · 2024-11-16
**Good paper, accept**

**Rating:** 7
**Confidence:** 4

**Review:**

The paper studies online imitation learning in an MDP. At each step, the learner must either take the optimal action, or query an oracle that supplies the optimal action (possibly with noise). The goal is to query the oracle as infrequently as possible.

The problem statement is clean and well-motivated, and the results are reasonably comprehensive. The most significant improvement over previous work in my opinion is that the results hold in an online setting, rather than requiring independent trajectories from the environment.

In terms of weaknesses, there are still large gaps between the upper and lower bounds for settings where the oracle is noisy. Also, the proof techniques rely heavily on existing methods in active learning, but I am not overly concerned about the lack of novelty, as I think the results fill a missing gap in the literature.

The authors should strongly consider providing more of the intuition behind the algorithms. While the first algorithm is straightforward enough, ReliableApprentice is difficult to parse, and chasing down the references did not help. Otherwise, the paper is quite well-written and easy to follow.

Also, I think that there is a missing conjecture? Conjecture 18 asks whether the query complexity can be reduced to match the lower bound in Theorem 12, but the lower bound in Theorem 9 has also not been matched, so why not make a similar conjecture about it?

**Paper Award:**

No

---

### Author Rebuttal · Authors · 2024-11-25

We thank all of the reviewers for their encouraging remarks and helpful comments and feedback.
We address a few comments raised in the reviews below:

Both reviewers Szof and P1mS ask about our conjecture regarding the lower bound being sharp for our Tsybakov-like condition, and whether we imagine a similar conjecture for the Massart condition.  Actually, we suspect a lower bound of eluder/Delta^2 should generally hold for the Massart condition, so that the upper bound is nearly sharp.  But proving this may require a new technique to handle the dependences due to the sequential aspects of the problem.  We will add a remark about this in the final version of the paper.

Regarding the specific example mentioned by reviewer P1mS (namely, the completely unstructured policy class), one can indeed show a |S|/Delta^2 lower bound, but the interpretation is less clear: i.e., in the proof of that lower bound, is |S| playing the role of log(|Pi|) or the role of the eluder dimension?

We thank reviewer Szof for the suggested improvements to readability.  We will add additional intuitive explanations of the algorithm in the final version of the paper.

Let us address reviewer GEdn's comment about novelty of results and analysis:

1. We note that the setting itself is natural and new to this work.

2. The analysis of disagreement-based active learning in MDP environments is new too.  Sure, the realizable case is straightforward.  But once noise is involved, we're in new territory (note we can't use traditional techniques like the disagreement coefficient or star number, since it isn't iid).  The key insight that made it all work is Lemma 13, which relates the number of disagreement points by a loss-constrained subclass to the eluder dimension and the loss constraint.  This lemma itself should be useful beyond this work.

3. Additionally, the formulation of the Tsybakov-like condition for this MDP-type setting is unlike anything we had seen before, and introduces some new interesting aspects to the analysis of disagreement-based active learning.

Regarding the question of reviewer GEdn about the use of Definition 10 in the proof of Lemma 16: Note that, in fact, Definition 10 requires that the stated inequality holds *simultaneously* for *every* $\tau > 0$ (not merely some *a priori* fixed $\tau$).  So, on this event, we are free to choose data-dependent values of $\tau$ and trust that the inequality still holds.

---

> ### Comment · Reviewer_Szof · 2024-11-26
>
> Thanks for your rebuttal. My score is unchanged.

---

> ### Comment · Reviewer_GEdn · 2024-11-28
> **Thank you for the rebuttal**
>
> Thank you for the detailed response. Thank you for your comments about the novelty of the analysis, I will keep my score for now.
> About the use of definition 10: Ok, thank you, this is my mistake. I didn't see that the event was about every $\tau$ simultaneously.

---

### Meta-Review · Area_Chair_ZGAJ · 2024-12-10

**Recommendation:** Accept
**Confidence:** 5

**Metareview:**

This paper introduces and studies a formulation of online apprenticeship learning in MDPs, in which the learner can either take an action by itself (and receive the corresponding reward and future-state feedback) or query an oracle that will return the optimal action for the current state (possibly in a noisy manner). The paper provides upper and lower bounds on the number of queries to the oracle as a function of the Eluder dimension of the policy class if the oracle is either noiseless or possesses Massart or Tsybakov noise.

This paper received three expert reviews. All of the reviewers generally like the paper for its formulation, algorithm and contributions, though one reviewer mentions concerns about technical novelty (particularly for the noiseless case, which the authors also acknowledge is straightforward). Multiple reviewers also mention the lack of tightness of bounds in the noisy case as a shortcoming, albeit also acknowledging that that is a challenging problem to resolve. Based on this, the authors are recommended to:
- discuss technical novelty of the analysis in the noisy case at length and explain why a) active learning tools from iid data cannot be applied, b) why their Lemma 13 could be of independent interest, e.g. in what other problem formulations?
- acknowledge more directly the limitations in the bounds not being tight and discuss what open problem would need to be resolved to address this,
- read carefully all reviewer comments on readability and address each of them,

in the camera-ready version.

**Paper Award:**

No